# Trajectory Bellman Residual Minimization:
# A Simple Value-Based Method for LLM Reasoning

**Yurun Yuan**[1]    **Fan Chen**[2]    **Zeyu Jia**[2]    **Alexander Rakhlin**[2*]    **Tengyang Xie**[1*]

[1]University of Wisconsin–Madison        [2]Massachusetts Institute of Technology
{yurun_yuan,tx}@cs.wisc.edu
{fanchen,zyjia,rakhlin}@mit.edu

## Abstract

Policy-based methods currently dominate reinforcement learning (RL) pipelines for large language model (LLM) reasoning, leaving value-based approaches largely unexplored. We revisit the classical paradigm of Bellman Residual Minimization and introduce Trajectory Bellman Residual Minimization (TBRM), an algorithm that naturally adapts this idea to LLMs, yielding a simple yet effective off-policy algorithm that optimizes a single trajectory-level Bellman objective using the model's own logits as $Q$-values. TBRM removes the need for critics, importance-sampling ratios, or clipping, and can operate with only one rollout per prompt. We prove convergence to the near-optimal KL-regularized policy from arbitrary off-policy data via an improved change-of-trajectory-measure analysis. Experiments on standard mathematical-reasoning benchmarks show that TBRM matches or surpasses policy-based baselines, like PPO and GRPO, with comparable or lower computational and memory overhead. Our results indicate that value-based RL might be a principled and efficient alternative for enhancing reasoning capabilities in LLMs. The codebase for TBRM is publicly available at `https://github.com/rlx-lab/TBRM`.

## 1  Introduction

Large language models (LLMs) have become the de-facto backbone for modern natural-language understanding and generation (Brown et al., 2020; Ouyang et al., 2022). While ever-larger pre-training corpora push the frontier of *knowledge*, high-value downstream usage increasingly hinges on *reasoning*: the capacity to carry out multi-step thinking, apply abstract rules to practical situations, and generalize from observed patterns to solve complex, structured problems. Reinforcement learning (RL) with verifiable, outcome-based rewards has emerged as a powerful paradigm for enhancing this reasoning capability in LLMs (Guo et al., 2025), especially for mathematical problem-solving where correctness can be objectively determined.

Recent advances in LLM post-training for mathematical reasoning have primarily employed policy-based variants—Proximal Policy Optimization (PPO; Schulman et al., 2017b) and Group Relative Policy Optimization (GRPO; Shao et al., 2024; Guo et al., 2025)—which optimize policies to maximize objective rewards that indicate successful task completion. These approaches leverage the clear evaluation criteria of mathematical tasks, where responses can be automatically verified as correct or incorrect without requiring human judgment.

Despite their empirical success, policy-based methods face several practical challenges. They typically require fresh on-policy rollouts from the current model, increasing computational demands. They often rely on additional components like critic models, advantage normalization, and clipping

---

*Corresponding authors.

mechanisms, adding complexity to implementation and tuning. Moreover, for token-level decisions, these methods (1) simplistically attribute outcome-based rewards (e.g., correctness of the entire response) to individual tokens, often assigning credit primarily to the final token, and (2) bootstrap advantages with truncated rollout horizons when additional critic models are used, potentially compromising effective credit assignment during training.

Classical RL offers a compelling alternative: *value-based* methods learn an action-value function $Q$ and act, for instance, greedily with respect to it. In the LLM setting, each token is an action, and the model's logits (that is, raw network outputs) naturally provide a parametric family for $Q$, under the KL-regularized RL framework (e.g., Schulman et al., 2017a). Despite this natural alignment and the inherent strengths of value-based methods for LLMs, their application to LLMs has been limited, with policy-based techniques being more prevalent. One potential reason, we conjecture, is the perceived difficulty in reconciling traditional iterative bootstrapping in value-based RL (e.g., the Q-learning family, Watkins and Dayan, 1992; Mnih et al., 2015) with the scale of LLM training, as iterative-style algorithms are typically less stable than their optimization-style counterparts (e.g., policy-based methods).

Our starting point is *Bellman Residual Minimization* (BRM; Schweitzer and Seidmann, 1985; Baird et al., 1995), a decades-old idea that fits the Q-function by directly minimizing its Bellman residual, designed for deterministic environments (which LLMs naturally are). By leveraging the recent theoretical advances in trajectory-level change of measure (Jia et al., 2025), we recognize that BRM can be lifted from the *step (token)* to the *trajectory* level: we square a single residual spanning the whole rollout and regress the model's logits onto it. This approach eliminates the aforementioned per-step-signal barrier, removes the need for critics, importance weights, and clipping, and provably maintains fully off-policy optimization due to the value-based nature of the algorithm. The resulting algorithm is *Trajectory BRM* (TBRM), which builds on classical BRM. Below we state our contributions, focused on theoretical analysis and extensive experiments on math reasoning tasks.

### 1.1 Our Results

1. **Algorithm.** Building explicitly on the classical idea of Bellman Residual Minimization, we present *TBRM*, a single-objective, off-policy algorithm that fits the trajectory-level Bellman residual using the LLM logits as $Q$-values. TBRM dispenses with critics, advantage estimates, importance ratios, or clipping, and can operate with only *one rollout per prompt*, while scaling effectively with multiple rollouts in practice.

2. **Theory.** We prove that, under standard realizability assumptions, the algorithm converges to the *optimal KL-regularized policy* even when training data are generated by *arbitrary* behavior policies in deterministic environments (such as LLMs). Our results build upon the recent change-of-trajectory-measure result of Jia et al. (2025). We significantly simplify that proof and improve the rate of convergence in terms of horizon factors. Overall, our results offer a theoretically grounded alternative to popular (yet ad-hoc) methods like GRPO.

3. **Experiments.** On six mathematical-reasoning benchmarks—namely AIME24/25, AMC23, MATH500, Minerva-Math, and OlympiadBench—TBRM performs on par with or better than PPO and GRPO baselines. Notably, TBRM achieves up to 30.5% accuracy on AIME24 with Qwen2.5-Math-7B. Compared to GRPO, it improves the average benchmark score by 1.3% absolute, while under comparable conditions to PPO, it achieves better performance with 22.5% less training time and 33% lower GPU memory. We further demonstrate that TBRM benefits from additional rollouts and the model learns emergent reasoning patterns, such as verification, backtracking, and decomposition, that align with human mathematical practice.

Collectively, our findings suggest that value-based approaches offer a compelling alternative to policy gradient methods especially for enhancing mathematical reasoning capabilities in LLMs. By recasting value learning at the trajectory level, TBRM provides a *principled, efficient, and theoretically grounded* approach for improving performance on mathematical reasoning tasks while dramatically reducing computational requirements.

## 2 Preliminaries

This section provides the necessary background for our work. We first review the fundamentals of KL-regularized RL (Section 2.1), then discuss prominent reinforcement learning algorithms applied to

large language models (Section 2.2), and finally introduce the autoregressive function approximation framework (Section 2.3) that serves as the foundation for our proposed approach.

## 2.1 KL-Regularized Reinforcement Learning

Reinforcement learning (RL) provides a framework for sequential decision-making problems where an agent interacts with an environment to maximize cumulative rewards. In the context of Markov Decision Processes (MDPs), which provide the theoretical foundation for RL, we consider an episodic finite-horizon framework. Formally, a horizon-$H$ episodic MDP $M = (H, \mathcal{S}, \mathcal{A}, \mathcal{P}, r, \rho)$ consists of a (potentially very large) state space $\mathcal{S}$, an action space $\mathcal{A}$, a probability transition function $\mathcal{P} : \mathcal{S} \times \mathcal{A} \to \Delta(\mathcal{S})$, a reward function $r : \mathcal{S} \times \mathcal{A} \to \mathbb{R}$, and an initial state distribution $\rho \in \Delta(\mathcal{S})$. The state space is typically layered such that $\mathcal{S} = \mathcal{S}_1 \cup \mathcal{S}_2 \cup \cdots \cup \mathcal{S}_H$, where $\mathcal{S}_h$ is the set of states reachable at step $h$. A policy $\pi : \mathcal{S} \to \Delta(\mathcal{A})$ maps states to distributions over actions and induces a distribution over trajectories $\tau = (s_1, a_1, \ldots, s_H, a_H)$ and rewards $(r_1, \ldots, r_H)$, where the initial state is sampled as $s_1 \sim \rho$, and for $h = 1, \ldots, H$: $a_h \sim \pi(s_h)$, $r_h = r(s_h, a_h)$, and $s_{h+1} \sim \mathcal{P}(s_h, a_h)$. We let $\mathbb{E}_{\tau \sim \pi}[\cdot]$ and $\mathbb{P}_{\tau \sim \pi}[\cdot]$ denote expectation and probability under this process, and $\mathbb{E}_\pi[\cdot]$ and $\mathbb{P}_\pi[\cdot]$ for brevity when $\tau$ is not explicitly mentioned.

For any policy $\pi$, we define the occupancy measures that characterize the probabilities of visiting states and selecting actions when following $\pi$. Specifically, the state occupancy measure $d^\pi(s_h) :=$ $\mathbb{P}_{\tau \sim \pi}[s_h \in \tau]$ represents the probability of visiting state $s_h \in \mathcal{S}_h$ under policy $\pi$. Similarly, the state-action occupancy measure $d^\pi(s_h, a_h) := \mathbb{P}_{\tau \sim \pi}[(s_h, a_h) \in \tau]$ gives the probability of the state-action pair $(s_h, a_h)$ occurring in a trajectory. We also define the trajectory occupancy measure $d^\pi(\tau) := \mathbb{P}_{\tau' \sim \pi}[\tau' = \tau]$, which is the probability of generating the exact trajectory $\tau$ when following policy $\pi$. It is important to distinguish between $d^\pi(\tau)$ and $\pi(\tau) := \prod_{(s_h, a_h) \in \tau} \pi(a_h \mid s_h)$, as they differ when the transition dynamics are stochastic.

In standard RL, the objective is to find a policy $\pi$ that maximizes the expected cumulative reward $J(\pi) = \mathbb{E}_{\tau \sim \pi}[r(\tau)]$, where $r(\tau) = \sum_{h=1}^H r(s_h, a_h)$. In many practical applications, particularly in the context of large language models, it is beneficial to incorporate a regularization term that encourages the learned policy to stay close to a reference policy $\pi_{\text{ref}}$. This leads to the KL-regularized RL objective (Ziebart et al., 2008; Ziebart, 2010; Neu et al., 2017; Ouyang et al., 2022)

$$J_\beta(\pi) = \mathbb{E}_{\tau \sim \pi}[r(\tau)] - \beta \cdot \mathbb{E}_{\tau \sim \pi}\left[\log \frac{\pi(\tau)}{\pi_{\text{ref}}(\tau)}\right] = \mathbb{E}_{\tau \sim \pi}\left[\sum_{h=1}^H \left(r(s_h, a_h) - \beta \log \frac{\pi(a_h|s_h)}{\pi_{\text{ref}}(a_h|s_h)}\right)\right],$$

where $\beta > 0$ is a regularization parameter that controls the strength of the penalty $D_{\text{KL}}(\pi\|\pi_{\text{ref}}) = \mathbb{E}_{\tau \sim \pi}\left[\log \frac{\pi(\tau)}{\pi_{\text{ref}}(\tau)}\right]$, known as the Kullback-Leibler divergence.

## 2.2 Reinforcement Learning Methods for Large Language Models

In this section, we briefly review popular reinforcement learning methods for large language models. For the sake of generality, we continue to use the terminology of MDPs (i.e., we use $s$ to represent state and $a$ to represent action). This terminology naturally encompasses the case of large language models in both single-turn and multi-turn interaction settings.

In the single-turn setting where $x \sim \rho$ denotes the input prompt and $y_1, y_2, \ldots, y_H$ denote the output tokens, we can define $s_1 := x$ and $s_h := (x, y_1, \ldots, y_{h-1})$ for $h > 1$, with $a_h := y_h$ for $h = 1, \ldots, H$. In the multi-turn setting, which consists of multiple interaction turns $(x^{(1)}, y_{1:H}^{(1)})$, $(x^{(2)}, y_{1:H}^{(2)})$, and so forth, we can adapt the transition function accordingly. Here, $y_{1:H}^{(i)}$ is a shorthand notation for the sequence of tokens $y_1^{(i)}, y_2^{(i)}, \ldots, y_H^{(i)}$ in the $i$-th turn. For instance, if a state-action pair $(s, a)$ contains the complete response for one turn (e.g., in a conversation with three or more turns), where $s = (x^{(1)}, y_{1:H}^{(1)}, x^{(2)}, y_{1:H-1}^{(2)})$ and $a = y_H^{(2)}$, the next state would transition to $s' = (x^{(1)}, y_{1:H}^{(1)}, x^{(2)}, y_{1:H}^{(2)}, x^{(3)})$, rather than simply concatenating the previous state and action as in the single-turn case.

**Proximal Policy Optimization (PPO).** PPO (Schulman et al., 2017b) introduces a clipped surrogate objective to constrain policy updates:

$$\mathcal{J}^{\text{PPO}}(\theta) = \mathbb{E}_{(s_h, a_h) \sim \pi_{\theta_{\text{old}}}}\left[\min\left(\frac{\pi_\theta(a_h|s_h)}{\pi_{\theta_{\text{old}}}(a_h|s_h)}\widehat{A}_h(s_h, a_h), \text{clip}\left(\frac{\pi_\theta(a_h|s_h)}{\pi_{\theta_{\text{old}}}(a_h|s_h)}, 1 - \varepsilon, 1 + \varepsilon\right)\widehat{A}_h(s_h, a_h)\right)\right],$$

where $\widehat{A}_h$ is the advantage estimate, and $\varepsilon$ is a hyperparameter. The advantage $\widehat{A}_h$ is typically computed using Generalized Advantage Estimation (GAE; Schulman et al., 2015): $\widehat{A}_h(s_h, a_h) =$

$\sum_l (\lambda)^l \delta_{t+l}$, where $\delta_h = r_h + V_\phi(s_{h+1}) - V_\phi(s_h)$ is the temporal difference error and $V_\phi$ is an estimate of the value function of the KL regularized reward $r(s_h, a_h) - \beta \log \frac{\pi(a_h|s_h)}{\pi_{\text{ref}}(a_h|s_h)}$.

For LLMs, PPO has been widely used for enhancing mathematical reasoning capabilities, where objective rewards signal the correctness of the model's solutions.

**Group Relative Policy Optimization (GRPO).** GRPO (Shao et al., 2024) is a policy-based method that, in practical implementations for LLMs like DeepSeek-R1, samples $G$ responses $o^{(1)}, \ldots, o^{(G)}$ for each prompt $x$ and computes advantages by normalizing rewards within each prompt group. In the MDP terminology above, this corresponds to: $s_1^{(i)} = x \sim \rho$, $o^{(i)} = (a_1^{(i)}, a_2^{(i)}, \ldots, a_{|o^{(i)}|}^{(i)})$, $s_h^{(i)} = (x, a_1^{(i)}, \ldots, a_{h-1}^{(i)})$, and $r(x, o^{(i)}) = \sum_h r(s_h^{(i)}, a_h^{(i)})$. The advantage for the $i$-th response $o^{(i)}$ (and implicitly for each token within that response) is computed as: $\widehat{A}^{(i)} = \frac{r(x, o^{(i)}) - \text{mean}(\{r(x, o^{(1)}), \ldots, r(x, o^{(G)})\})}{\text{std}(\{r(x, o^{(1)}), \ldots, r(x, o^{(G)})\})}$, where $r(x, o^{(i)})$ is the outcome for response $o^{(i)}$ to prompt $x$ as we defined above. This response-level advantage $\widehat{A}^{(i)}$ is then used to replace the step-wise advantage function $\widehat{A}_h(s_h, a_h)$ in the PPO objective $\mathcal{J}^{\text{PPO}}$, but then GRPO objective accommodates the KL-regularization at the end:

$$\mathcal{J}^{\text{GRPO}}(\theta) = \mathbb{E}_{x \sim \rho, \{o^{(i)}\}_{i=1}^G \sim \pi_{\theta_{\text{old}}}(\cdot|x)} \left[ \frac{1}{G} \sum_{i=1}^{G} \frac{1}{|o_i|} \sum_{h=1}^{|o_i|} \left\{ \right. \right.$$

$$\left. \left. \min \left[ \frac{\pi_\theta(a_h^{(i)} \mid s_h^{(i)})}{\pi_{\theta_{\text{old}}}(a_h^{(i)} \mid s_h^{(i)})} \widehat{A}^{(i)}, \text{clip}\left( \frac{\pi_\theta(a_h^{(i)} \mid s_h^{(i)})}{\pi_{\theta_{\text{old}}}(a_h^{(i)} \mid s_h^{(i)})}, 1 - \varepsilon, 1 + \varepsilon \right) \widehat{A}^{(i)} \right] - \beta D_{\text{KL}}(\pi \| \pi_{\text{ref}}) \right\} \right].$$

The normalization by mean and standard deviation is intended to stabilize training by reducing variance. GRPO is often considered a simpler alternative to PPO for post-training LLMs. This is partly because PPO typically involves training a separate critic network and incorporates more complex mechanisms for policy updates. In the context of LLMs, the full complexity of PPO might not always be necessary, leading to the adoption of more streamlined policy gradient methods like GRPO.

## 2.3 Autoregressive Function Approximation

Having established the principles of KL-regularized RL and reviewed current RL methods for LLMs, we now introduce the key formulation that bridges these concepts: autoregressive function approximation. This framework allows us to naturally parameterize value functions and policies using autoregressive models like LLMs, which is central to our proposed method.

Note that KL-regularized RL has been widely studied in classical RL literature (e.g., Schulman et al., 2017a; Nachum et al., 2017; Haarnoja et al., 2018) and the similar idea of autoregressive function approximation has also been introduced by Guo et al. (2022). This subsection should be viewed as a discussion of preliminary background and unified notations that will enable our proposed approach.

Given a reference model $\pi_{\text{ref}}$, we first define the following modified reward function

$$R_\beta(s_h, a_h) = \frac{r(s_h, a_h)}{\beta} + \log \pi_{\text{ref}}(a_h \mid s_h), \tag{1}$$

and, therefore, the original KL-regularized RL objective can be rewritten as: $J_\beta(\pi) = \beta \cdot \mathbb{E}_{\tau \sim \pi}[R_\beta(\tau) - \log \pi(\tau)] = \beta \cdot \mathbb{E}_{\tau \sim \pi}\left[ \sum_{h=1}^{H} (R_\beta(s_h, a_h) - \log \pi(a_h \mid s_h)) \right]$. Here, KL-regularization is equivalent to entropy regularization. However, we will continue to use the KL-regularization terminology throughout the remainder of the paper for consistency.

The optimal policy for the above objective, denoted $\pi_\beta^\star = \arg\max_\pi J_\beta(\pi)$, has a closed-form solution that takes the form of a softmax distribution,

$$\pi_\beta^\star(a_h \mid s_h) \propto \pi_{\text{ref}}(a_h \mid s_h) \exp\left( Q_r^\star(s_h, a_h) / \beta \right), \quad \text{or} \quad \pi_\beta^\star(a_h \mid s_h) \propto \exp\left( Q_{R_\beta}^\star(s_h, a_h) \right),$$

where $Q_r^\star$ and $Q_{R_\beta}^\star$ are the optimal action-value functions for the original reward $r$ and transformed reward functions $R_\beta$, respectively.

We now formalize the Bellman operators under the shifted reward function $R_\beta$, and discuss the key properties induced by the KL-regularization. For a given policy $\pi$ and any Q-function $Q$, we define the Bellman operator as

$$(\mathcal{T}_\beta^\pi Q)(s_h, a_h) := R_\beta(s_h, a_h) + \mathbb{E}_{s_{h+1} \sim \mathcal{P}(\cdot|s_h, a_h), a_{h+1} \sim \pi(\cdot|s_{h+1})} \left[ Q(s_{h+1}, a_{h+1}) - \log \pi(a_{h+1} \mid s_{h+1}) \right].$$

Throughout this paper, we will use $V_Q(s) := \log \sum_{a \in \mathcal{A}} e^{Q(s,a)}$ to denote the softmax of the given $Q$. With this, the Bellman optimality operator becomes

$$(\mathcal{T}_\beta Q)(s_h, a_h) := R_\beta(s_h, a_h) + \mathbb{E}_{s_{h+1} \sim \mathcal{P}(\cdot | s_h, a_h)} \Big[ \underbrace{\max_{\pi \in \Delta(\mathcal{A})} \mathbb{E}_{a_{h+1} \sim \pi(\cdot | s_{h+1})} [Q(s_{h+1}, a_{h+1}) - \log \pi(a_{h+1} | s_{h+1})]}_{= \log \sum_{a \in \mathcal{A}} \exp(Q(s_{h+1}, a)) =: V_Q(s_{h+1})} \Big]$$

$$= R_\beta(s_h, a_h) + \mathbb{E}_{s_{h+1} \sim \mathcal{P}(\cdot | s_h, a_h)} [V_Q(s_{h+1})].$$

With the above definitions, the optimal policy takes the form of the following Boltzmann distribution:

$$\pi_Q(\cdot | s_h) := \underset{\pi \in \Delta(\mathcal{A})}{\arg\max} \, \mathbb{E}_{a_h \sim \pi(\cdot | s_h)} [Q(s_h, a_h) - \log \pi(a_h | s_h)] = e^{Q(s_h, \cdot) - V_Q(s_h)}.$$

We can further define the KL-regularized value functions for a given policy $\pi$:

$$Q_\beta^\pi(\widetilde{s}_h, \widetilde{a}_h) := \mathbb{E}_\pi \Big[ R_\beta(s_h, a_h) + \sum_{h'=h+1}^{H} (R_\beta(s_{h'}, a_{h'}) - \log \pi(a_{h'} | s_{h'})) \,\Big|\, (s_h, a_h) = (\widetilde{s}_h, \widetilde{a}_h) \Big],$$

$$V_\beta^\pi(\widetilde{s}_h) := \mathbb{E}_\pi \Big[ \sum_{h'=h}^{H} (R_\beta(s_{h'}, a_{h'}) - \log \pi(a_{h'} | s_{h'})) \,\Big|\, s_h = \widetilde{s}_h \Big] = \mathbb{E}_{a_h \sim \pi(\cdot | \widetilde{s}_h)} [Q_\beta^\pi(\widetilde{s}_h, a_h) - \log \pi(a_h | \widetilde{s}_h)].$$

The corresponding optimal value functions are $Q_\beta^\star := Q_\beta^{\pi_\beta^\star}$ and $V_\beta^\star := V_\beta^{\pi_\beta^\star}$. Here, $Q_\beta^\pi$ and $Q_\beta^\star$ are also the fixed points of $\mathcal{T}_\beta^\pi$ and $\mathcal{T}_\beta$, respectively.

For autoregressive function approximation architectures, such as large language models, we can directly leverage logits to parameterize $Q$, $V_Q$, and $\pi_Q$ as follows. Let $\theta$ be the model weights. Define

$$\boxed{Q_\theta(s, a) := \mathsf{logit}_\theta(s, a), \quad V_\theta(s) := \mathsf{softmax} \circ \mathsf{logit}_\theta(s, \cdot), \quad \log \pi_\theta(a | s) := Q_\theta(s, a) - V_\theta(s),} \quad (2)$$

where $\mathsf{softmax} \circ \mathsf{logit}_\theta(s, \cdot) := \log \sum_{a \in \mathcal{A}} \exp(\mathsf{logit}_\theta(s, a))$.[2] While we assumed here temperature to be 1 for simplicity, any temperature can be incorporated by appropriately scaling the reward Eq. (1).

## 3 Trajectory Bellman Residual Minimization

In this section, we introduce our main algorithm, **Trajectory Bellman Residual Minimization (TBRM)**, designed specifically for large language models problems. As we discussed in Section 2.1, the transition dynamics of large language models can be viewed as deterministic. *For the remainder of this section, we will apply the autoregressive function approximation defined in Eq. (2) and assume deterministic transition dynamics.*

Recall that Bellman error $Q(s_h, a_h) - (\mathcal{T}_\beta Q)(s_h, a_h)$ over state-action pairs $(s_h, a_h)$ is employed as the proxy for controlling the performance $J_\beta(\pi_Q)$ of $\pi_Q$. Minimizing the square of Bellman error on $(s_h, a_h)$ in deterministic MDPs is equivalent for minimizing the square of Bellman residual $Q(s_h, a_h) - R_\beta(s_h, a_h) - V_Q(s_{h+1})$ given the $(s_h, a_h, s_{h+1})$ tuple. This leads to the classical Bellman residual minimization objective (BRM; Schweitzer and Seidmann, 1985; Baird et al., 1995), which we expand using the definition of $R_\beta$ and autoregressive function approximation in Eq. (2):

$$\mathcal{L}_\mathcal{D}^{\mathsf{BRM}}(\theta) = \frac{1}{|\mathcal{D}|} \sum_{\tau \in \mathcal{D}} \sum_{h=1}^{H} \Big( \mathsf{logit}_\theta(s_h, a_h) - (\mathcal{T}_\beta \mathsf{logit}_\theta)(s_h, a_h) \Big)^2 \quad (3)$$

$$= \frac{1}{|\mathcal{D}|} \sum_{\tau \in \mathcal{D}} \sum_{h=1}^{H} \Big( \mathsf{logit}_\theta(s_h, a_h) - \frac{r(s_h, a_h)}{\beta} - \log \pi_{\mathsf{ref}}(s_h, a_h) - \log \sum_{a \in \mathcal{A}} \exp(\mathsf{logit}_\theta(s_{h+1}, a)) \Big)^2.$$

Here $\mathcal{D}$ denotes data which can be either purely offline or updated online as a replay buffer. In the context of LLMs, directly minimizing BRM may not be possible because the token-level reward signal is either unavailable (e.g., if we assign the outcome reward to the final token) or very sparse. Crucially, minimizing the square of per-step Bellman error as in $\mathcal{L}_\mathcal{D}^{\mathsf{BRM}}(\theta)$ is sufficient but not necessary for maximization of $J_\beta(\pi_\theta)$. Indeed, a weaker control of Bellman errors over certain distributions is sufficient for optimizing $J_\beta(\pi_\theta)$ (see, e.g., Xie and Jiang (2020) or Corollary 4). As we prove below (Section 3.1), it is sufficient to instead consider a trajectory-based variant of BRM,

$$\mathcal{L}_\mathcal{D}^{\mathsf{TBRM}}(\theta) = \frac{1}{|\mathcal{D}|} \sum_{\tau \in \mathcal{D}} \Big( \sum_{h=1}^{H} \mathsf{logit}_\theta(s_h, a_h) - (\mathcal{T}_\beta \mathsf{logit}_\theta)(s_h, a_h) \Big)^2$$

---

[2]Here, $\mathsf{softmax}$ denotes the $\mathsf{log\text{-}sum\text{-}exp}$ operator for notational convenience, rather than the vocabulary softmax layer used to produce token probabilities in language models.

$$= \frac{1}{|\mathcal{D}|} \sum_{\tau \in \mathcal{D}} \left( \mathsf{logit}_\theta(s_1, a_1) - \frac{r(\tau)}{\beta} - \log \pi_{\mathsf{ref}}(\tau) + \sum_{h=2}^{H} \log \pi_\theta(a_h \mid s_h) \right)^2, \qquad (4)$$

where the second equality used the autoregressive function approximation, i.e., $\log \pi_\theta(a_h \mid s_h) = \mathsf{logit}_\theta(a_h \mid s_h) - \log \sum_{a \in \mathcal{A}} \exp(\mathsf{logit}_\theta(s_h, a))$, to simplify the expression. An immediate benefit of TBRM is that we only require the trajectory outcome $r(\tau)$, rather than the process reward $r(s_h, a_h)$ as in BRM, and credit assignment will be (provably) carried out in the learning procedure itself. In our experiments, we update $\mathcal{D}$ online, and Algorithm 1 represents the exact implementation we used in Section 4.

---

**Algorithm 1** Trajectory Bellman Residual Minimization (TBRM)

---

**input:** Task prompt dataset $\mathcal{D}_{\mathsf{task}}$, reward function $r$, reward scaling coefficient $\beta$, reference policy $\pi_{\mathsf{ref}}$ with parameter $\theta_{\mathsf{ref}}$, and number of iterations $T$.

1: Initialize $\theta \leftarrow \theta_{\mathsf{ref}}$.
2: **for** $t = 1$ to $T$ **do**
3:      Sample a batch $\mathcal{D}_{\mathsf{task}}^t \subset \mathcal{D}_{\mathsf{task}}$.
4:      For each question $q \in \mathcal{D}_{\mathsf{task}}^t$, sample a trajectory $\tau$ from policy $\pi_\theta$ with initial state $s_1 = q$, and collect these trajectories into dataset $\mathcal{D}_t$.
5:      Update $\theta$ via gradient descent to minimize $\mathcal{L}_{\mathcal{D}_t}^{\mathsf{TBRM}}(\theta)$ as defined in Eq. (4):

$$\nabla_\theta \frac{1}{|\mathcal{D}_t|} \sum_{\tau \in \mathcal{D}_t} \left( \mathsf{logit}_\theta(s_1, a_1) - \frac{r(\tau)}{\beta} - \log \pi_{\mathsf{ref}}(\tau) + \sum_{h=2}^{H} \log \pi_\theta(a_h \mid s_h) \right)^2,$$

     where $\mathsf{logit}_\theta(s_1, a_1)$ is the raw logit of the first output token.
6: **end for**
7: **return** $\pi_\theta$.

---

As we formally prove in Section 3.1, TBRM is a fully *off-policy* algorithm with a directly optimizable objective, and it provably converges to a near-optimal policy with any off-policy data (though the degree of off-policyness may affect sample efficiency, Eq. (6)). In contrast, policy-based counterparts are usually on-policy in nature: policy-gradient-based algorithms (like REINFORCE) require to sample new trajectories in an on-policy manner. PPO-like algorithms require on-policy actions for their actor components but optimize a surrogate loss instead. PPO's critic update also requires on-policy rollouts. In contrast, the off-policy nature of TBRM removes the need for additional techniques such as importance sampling ratios, clipping, critic models, or (multiple) on-policy rollouts.

Readers may question: *why hasn't this simple variant of BRM received attention in the literature?* We conjecture that TBRM has been suspected to suffer from the curse of horizon, at least from the theoretical perspective. In the theory of offline RL (e.g., Chen and Jiang, 2019; Xie and Jiang, 2020), a key technique is to control the expected Bellman error $|\mathbb{E}_\pi[\sum_h((\mathcal{T}_\beta Q)(s_h, a_h) - Q(s_h, a_h)))]|$ on a certain (unavailable) distribution $d^\pi$ by instead minimizing the per-step squared Bellman error $\sum_h \mathbb{E}_\mu[((\mathcal{T}_\beta Q)(s_h, a_h) - Q(s_h, a_h))^2]$ on the data distribution $d^\mu$ generated by $\mu$. This step only incurs the cost of the state-wise distribution-shift $\frac{d^\pi(s_h, a_h)}{d^\mu(s_h, a_h)}$. When it comes to trajectory-level data, minimization of the square of expected Bellman error $\mathbb{E}_\mu[(\sum_h[(\mathcal{T}_\beta Q)(s_h, a_h) - Q(s_h, a_h)])^2]$, as in TBRM, would appear to incur the trajectory-level distribution-shift cost $\Pi_h \frac{\pi(a_h|s_h)}{\mu(a_h|s_h)}$, and thus possibly cause an exponential blow-up with horizon $H$ compared to the state-wise case. However, the recent theoretical results (Jia et al., 2025) challenge this conventional wisdom and indicate that the Markov property can be the key to avoiding trajectory-level distribution-shift when conducting trajectory-level change of measure. In Section 3.1, we formally prove that TBRM indeed only incurs state-wise distribution-shift regardless of its trajectory-level objective, and show that TBRM can efficiently converge to a near-optimal policy with finite-sample analysis.

**Comparison with other related algorithms.** We note that algorithms with similar structure to TBRM have been derived previously from diverse perspectives in both deep RL (Haarnoja et al., 2017; Schulman et al., 2017a; Nachum et al., 2017; Haarnoja et al., 2018) and LLM applications (Guo et al., 2022; Ethayarajh et al., 2024; Team et al., 2025; Ji et al., 2024; Wang et al., 2024). This convergence is unsurprising, as TBRM and related algorithms fundamentally aim to minimize Bellman error, albeit through different formulations and optimization approaches. However, to the

best of our knowledge, TBRM is the only optimization algorithm (rather than iterative ones like Q-learning; Appendix F demonstrates the benefit of optimization over iteration) that requires only one rollout per prompt among all of these approaches. Appendix E provides a detailed comparison of TBRM with other related algorithms. The present paper formally establishes finite-sample guarantees for TBRM.

## 3.1 Theoretical Analysis of TBRM

We use $\Theta$ to denote the parameter space, equipped with norm $\|\cdot\|$. We assume the following standard realizability condition, which can be relaxed to hold approximately (see, e.g., Cheng et al., 2022).

**Assumption 1** (Realizability). *There exists $\theta^\star \in \Theta$ such that $Q_{\theta^\star} = Q^\star$.*

**Motivation.** We first show that $\theta^\star$ is the population minimizer of the TBRM loss (4) through Bellman equation. Under the parametrization (2), $Q_{\theta^\star}(s, a) = \text{logit}_{\theta^\star}(s, a)$ is the optimal soft Q-function for the transformed reward function $R_\beta(s, a) = \frac{r(s,a)}{\beta} + \log \pi_{\text{ref}}(a \mid s)$, and the optimal value function is given by $V_{\theta^\star}(s) = Q_{\theta^\star}(s, a) - \log \pi_{\theta^\star}(a \mid s)$. Therefore, the Bellman equation becomes (deterministically for a trajectory $\tau$ drawn from the MDP)

$$Q_{\theta^\star}(s_h, a_h) = R_\beta(s_h, a_h) + V_{\theta^\star}(s_{h+1}) = R_\beta(s_h, a_h) + Q_{\theta^\star}(s_{h+1}, a_{h+1}) - \log \pi_{\theta^\star}(a_{h+1} \mid s_{h+1}).$$

Then, summing over $h = 1, 2, \cdots, H - 1$ for any admissible trajectory $\tau$, we have

$$0 \equiv \sum_{h=1}^{H-1} \left[ Q_{\theta^\star}(s_h, a_h) - R_\beta(s_h, a_h) - Q_{\theta^\star}(s_{h+1}, a_{h+1}) + \log \pi_{\theta^\star}(a_{h+1} \mid s_{h+1}) \right]$$

$$= \text{logit}_{\theta^\star}(s_1, a_1) - R(\tau) + \sum_{h=2}^{H} \log \pi_{\theta^\star}(a_h \mid s_h),$$

where $R(\tau) = \frac{r(\tau)}{\beta} + \log \pi_{\text{ref}}(\tau)$ is the trajectory transformed reward. Hence, it holds that $\mathcal{L}_{\mathcal{D}}^{\text{TBRM}}(\theta^\star) \equiv 0$ deterministically, and any approximate minimizer of the loss $\mathcal{L}_{\mathcal{D}}^{\text{TBRM}}$ must also attain low trajectory Bellman residual.

The analysis above establishes a necessary condition for the optimal soft Q-function $Q_{\theta^\star}(s, a)$ or $\text{logit}_{\theta^\star}(s, a)$. Beyond this, the sub-optimality of a policy $\pi_\theta$ can also be related to the trajectory Bellman residual: through our analysis in Appendix B, we can show that for any $\theta \in \Theta$,

$$J_\beta(\pi^\star) - J_\beta(\pi_\theta) \leq 2\beta \max_{\pi \in \{\pi^\star, \pi_\theta\}} \left| \mathbb{E}_\pi \left[ \text{logit}_\theta(s_1, a_1) - R(\tau) + \sum_{h=2}^{H} \log \pi_\theta(a_h \mid s_h) \right] \right|.$$

Therefore, it remains to relate the expected trajectory Bellman residual under the off-policy distribution induced by $\mu$ and any policy $\pi$ through change-of-trajectory-measure.

**Change-of-trajectory-measure.** A key to our analysis is the following improved version of the *change-of-trajectory-measure lemma* (Jia et al., 2025). Let $\chi^2(P \parallel Q) = \mathbb{E}_P[dP/dQ] - 1$ be the $\chi^2$-divergence. Let $d_h^\pi(\cdot), d_h^\mu(\cdot) \in \Delta(\mathcal{S}_h \times \mathcal{A})$ denote the occupancy measures of Markovian policies $\pi$ and $\mu$.

**Lemma 1** (Change-of-Trajectory-Measure Lemma). *Given an MDP $M = (H, \mathcal{S}, \mathcal{A}, \mathcal{P}, r, \rho)$ and a policy $\pi$,*

$$\sup_f \frac{\left( \mathbb{E}_\pi \left[ \sum_{h=1}^{H} f(s_h, a_h) \right] \right)^2}{\mathbb{E}_\mu \left[ \left( \sum_{h=1}^{H} f(s_h, a_h) \right)^2 \right]} \leq 1 + \sum_{h=1}^{H} \chi^2 \left( d_h^\pi \parallel d_h^\mu \right), \tag{5}$$

*where the supremum is over all measurable functions $f : \mathcal{S} \times \mathcal{A} \to \mathbb{R}$.*

The proof of Lemma 1 significantly simplifies the one in Jia et al. (2025), as shown in Appendix C. As a direct corollary, the RHS of Eq. (5) can be further upper bounded by $H \cdot \max_{h,s,a} \frac{d_h^\pi(s_h, a_h)}{d_h^\mu(s_h, a_h)}$, improving upon Jia et al. (2025) by a factor of $H^2$.

Our main result will be stated in terms of the following *concentrability coefficient* of the data collection policy $\mu$:

$$C_{\text{conc}}(\mu) := 1 + \max_{\theta \in \Theta} \max_{h \in [H]} \chi^2 \left( d_h^{\pi_\theta} \parallel d_h^\mu \right), \tag{6}$$

a notion weaker than the commonly-studied $L_\infty$-concentrability $C_{\text{conc},\infty}(\mu) := \max_{\pi,h,s,a} \frac{d_h^\pi(s,a)}{d_h^\mu(s,a)}$.

The proof of the following theorem is deferred to Appendix D.

**Theorem 2.** *Suppose $\widehat{\theta}$ is a parameter that satisfies $\mathcal{L}_\mathcal{D}^{\text{TBRM}}(\widehat{\theta}) - \inf_{\theta \in \Theta} \mathcal{L}_\mathcal{D}^{\text{TBRM}}(\theta) \leq \varepsilon_{\text{opt}}$, data $\mathcal{D}$ are i.i.d. according to $\mu$, and Assumptions 1, 2 and 3 hold. Then, with high probability, it holds that*

$$J_\beta(\pi^\star) - J_\beta(\pi_{\widehat{\theta}}) \leq \widetilde{O}\left( \sqrt{H C_{\text{conc}}(\mu) \left( \beta^2 \varepsilon_{\text{opt}} + \frac{H^2 \dim(\Theta)}{|\mathcal{D}|} \right)} \right),$$

*where $\dim(\Theta)$ is the measure of the dimension of $\Theta$ defined in Assumption 3.*

## 4 Experiments

In this section, we present experiments to evaluate the performance of TBRM on reasoning tasks. We compare TBRM against two policy-based methods: GRPO and PPO. The codebase for the experiments is publicly available at `https://github.com/rlx-lab/TBRM`.

### 4.1 Experiment Setup

**Datasets and models.** We train our models using the prompt set from DAPO (Yu et al., 2025, Apache license 2.0), which comprises approximately 17.4k math problems sourced from the AoPS[3] website and official competition homepages. All problems are standardized to have integer answers. To demonstrate the generality of our method across model scales, we conduct experiments using Qwen2.5-Math-1.5B and Qwen2.5-Math-7B.

**Evaluation.** We assess the models' reasoning abilities on several standard math benchmarks: AIME24, AIME25, AMC23, MATH500 (Hendrycks et al., 2021), Minerva Math (Lewkowycz et al., 2022), and OlympiadBench (He et al., 2024). For MATH500, Minerva Math, and OlympiadBench, we generate a single response per problem and report the overall accuracy, denoted as **Avg@1**. For the smaller benchmarks AIME24, AIME25, and AMC23, where performance can fluctuate due to limited data, we generate 32 responses per problem and average the accuracies to mitigate the intrinsic randomness of LLM outputs; this metric is denoted as **Avg@32**. Responses are sampled with temperature 0 for **Avg@1** and temperature 1.0 for **Avg@32**. We employ Math-Verify (Kydlíček, 2025, Apache-2.0 license) as the verifier.

**Implementation details.** We implement our methods and baselines using the VERL framework (Sheng et al., 2024, Apache-2.0 license), following most of the recommended hyperparameter settings for GRPO and PPO. To balance performance and efficiency, we use a prompt batch size of 128 and a response length of 2048 tokens per training step. For PPO, we generate one response per prompt ($n = 1$), while for GRPO, which requires multiple rollouts, we generate four responses ($n = 4$). On TBRM, we experiment with both settings ($n = 1$ and $n = 4$). All responses are sampled with a temperature of 1.0. For TBRM, we set $\beta = 0.002$ across all experiments. All models are trained for the same number of steps. More details of our implementation can be found in Appendix G.

### 4.2 Main Results

The effectiveness of our algorithm is demonstrated in Table 1. Across six challenging math benchmarks, TBRM consistently matches or surpasses its comparable baselines. Specifically, with a single rollout per prompt, TBRM$_{n=1}$ achieves higher accuracies than PPO$_{n=1}$ on most benchmarks and matches the performance of GRPO$_{n=4}$, despite the latter using four times as many samples during training. Notably, our Qwen2.5-Math-1.5B-based model attains 13.2% accuracy on AIME24, outperforming both the Qwen2.5-Math-7B base model and the 1.5B GRPO$_{n=4}$ model. When increasing the number of sampled responses to four, TBRM$_{n=4}$ surpasses baselines by a larger margin. On AIME24, our 1.5B model reaches 14.3% accuracy, while the 7B model further advances to 30.5%, exceeding GRPO$_{n=4}$ by 1.6%. Additional results with more rollouts are presented in Appendix H.5.

---

[3] `https://artofproblemsolving.com/`

| Method | AIME24 | AIME25 | AMC23 | MATH500 | Minerva Math | OlympiadBench |
|--------|--------|--------|-------|---------|--------------|---------------|
| | Avg@32 | Avg@32 | Avg@32 | Avg@1 | Avg@1 | Avg@1 |
| **Qwen2.5-Math-1.5B** | 5.0 | 1.9 | 24.9 | 63.4 | 16.5 | 30.8 |
| PPO $n=1$ | 11.4 | 4.5 | 46.6 | **72.2** | 26.8 | 36.0 |
| TBRM $n=1$ | 13.2 | 5.6 | 48.6 | **72.2** | 27.2 | 35.7 |
| GRPO $n=4$ | 13.0 | **7.1** | 49.9 | 71.2 | 28.7 | **37.5** |
| TBRM $n=4$ | **14.3** | 6.9 | **52.0** | **72.2** | **30.5** | 36.1 |
| **Qwen2.5-Math-7B** | 10.6 | 2.8 | 31.6 | 67.4 | 13.2 | 29.3 |
| PPO $n=1$ | 25.4 | **13.2** | 63.4 | 76.4 | 33.8 | 39.3 |
| TBRM $n=1$ | 24.1 | **13.2** | 63.4 | 78.6 | **36.4** | 41.5 |
| GRPO $n=4$ | 28.9 | 10.7 | 66.8 | **79.8** | 36.0 | 42.5 |
| TBRM $n=4$ | **30.5** | 13.1 | **68.4** | **79.8** | **36.4** | **44.1** |

Table 1: Performance of various methods on math benchmarks, where $n$ denotes the number of responses sampled per prompt during training. For each benchmark, the highest accuracy across all methods is bolded, and the highest accuracy among methods with $n=1$ is underscored.

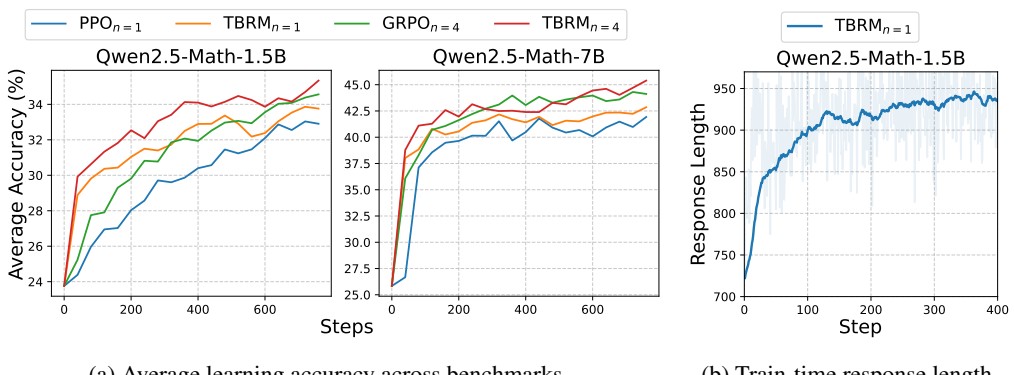

(a) Average learning accuracy across benchmarks     (b) Train-time response length.

Figure 1: Figure 1(a) shows average learning accuracy across benchmarks for PPO, GRPO, and TBRM. Per-benchmark results can be found in Appendix H.1. Figure 1(b) demonstrates model's ability to engage in extended reasoning at test time with R1 template.

Figure 1(a) illustrates the step-wise average performance across all benchmarks for TBRM and baseline methods. While all approaches demonstrate improved reasoning with increased training data, TBRM consistently exhibits a superior convergence rate and achieves higher absolute performance than its counterparts. Notably, TBRM$_{n=4}$ attains the highest performance throughout nearly the entire training duration. Futhermore, TBRM$_{n=1}$ outperforms PPO$_{n=1}$ and performs similarly to GRPO$_{n=4}$, especially on 1.5B model, with only a mild gap in their results.

### 4.3 Training Dynamics and Performance Analysis

**Reward.** We present the training reward curves in Figure 2(a), which shows that TBRM achieves comparable reward levels to its baselines. Furthermore, TBRM demonstrates a faster convergence rate during early training. This is particularly evident with the 1.5B model, where TBRM attains significantly higher rewards than PPO and GRPO.

**Response length.** Prior work has shown that reinforcement learning can enhance a model's ability to solve increasingly complex reasoning tasks by leveraging extended test-time computation, as reflected in progressively longer responses during training (Guo et al., 2025; Zeng et al., 2025; Liu et al., 2025). We find that TBRM exhibits a similar capability. Specifically, we adopt the prompt template from DeepSeek-R1 (Guo et al., 2025) and apply TBRM training to Qwen2.5-Math-1.5B. Following previous studies (Zeng et al., 2025), we include only responses that terminate under normal conditions—excluding those truncated due to length limits—as truncated outputs often suffer from repetition and incompleteness. As illustrated in Figure 1(b), TBRM encourages the model to explore

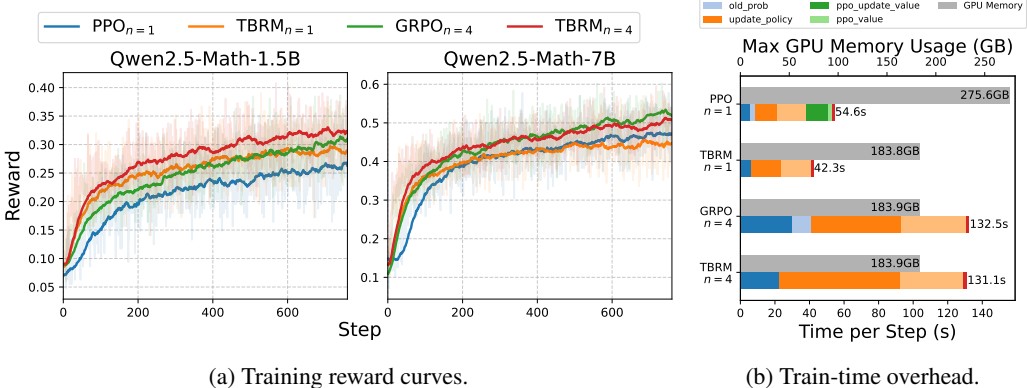

(a) Training reward curves.                    (b) Train-time overhead.

Figure 2: Figure 2(a) demonstrates the increment of rewards during training. Figure 2(b) is a comparison of maximal GPU memory consumption and per-step time cost across different methods with base model Qwen2.5-Math-7B. Time cost is segmented into key partitions, with time partition labels defined in Appendix G.2 (Appendix G.2).

and refine its reasoning more deeply over time, enabling models to take advantage of extended computation at test time to improve reasoning performance.

**Training efficiency.**    TBRM offers substantial implementation simplicity relative to existing approaches. Specifically, it eliminates the need for critic models $V_\phi$, as required by PPO, and avoids the necessity of sampling multiple responses per prompt ($n > 1$), as in GRPO. Moreover, due to its fully off-policy nature, TBRM does not require multiple updates per training step. We conduct experiments using Qwen2.5-Math-7B under consistent training conditions (see Appendix G.2 for details) and report the peak GPU memory usage and wall-clock time per training step for each method in Figure 2(b). Overall, TBRM exhibits matched or lower resource consumption compared to its counterparts. When $n = 1$, TBRM$_{n=1}$ uses 33.3% less GPU memory than PPO$_{n=1}$ and achieves a $1.3\times$ speedup. For $n = 4$, TBRM$_{n=4}$ demonstrates comparable resource usage to GRPO$_{n=4}$. Notably, TBRM$_{n=1}$ achieves a $3.1\times$ training speedup relative to GRPO$_{n=4}$, while yielding similar performance despite sampling only a single response—GRPO$_{n=4}$ outperforms TBRM$_{n=1}$ by only $0.80\%$ with the 1.5B model and $1.27\%$ with the 7B model on average across math benchmarks.

**Extended Analysis.**    We further analyze the responses generated by the TBRM models and identify several notable reasoning patterns, including verification, backtracking, decomposition, and enumeration. Illustrative examples of these patterns are provided in Appendix H.2. In Appendix H.3, we compare TBRM with the classical, token-level BRM formulation and show that directly applying BRM to LLMs leads to unstable training and reward collapse, underscoring the importance of the trajectory-level design. In Appendix H.4, we evaluate TBRM on a suite of non-mathematical reasoning tasks, demonstrating its ability to generalize beyond the mathematical domain.

## 5   Conclusion

In this paper, we present TBRM, a simple yet effective value-based RL algorithm for enhancing LLM reasoning capabilities. TBRM operates efficiently with just a single rollout per prompt and employs a lightweight optimization objective, eliminating the need for critics, importance-sampling ratios, or clipping mechanisms that are commonly required in policy-based approaches. Our theoretical analysis demonstrates that TBRM is guaranteed to converge to a near-optimal policy using off-policy data, while our empirical results show its effectiveness across standard mathematical reasoning benchmarks. We hope this work may contribute to expanding interest in value-based approaches for LLM reasoning, potentially complementing the policy-based algorithms that currently dominate LLM post-training methods.

## Acknowledgements

We acknowledge support of the Simons Foundation and the NSF through awards DMS-2031883 and PHY-2019786, ARO through award W911NF-21-1-0328, and the DARPA AIQ award.

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

# Appendix

## A Related Works

**Value-based RL.** Value-based methods are arguably the oldest and most widely studied concepts of reinforcement learning (RL) algorithms (Bellman, 1957; Samuel, 1959). They seek to learn an approximation of the optimal state-action-value function $Q^\star$ and act greedily with respect to it, in contrast to policy-gradient methods that directly optimize a parameterized policy. Early works such as Q-learning (Watkins and Dayan, 1992), SARSA (Sutton et al., 1998), Approximate Dynamic Programming (Bertsekas and Tsitsiklis, 1996) established the foundations, while the successive studies introduced function approximation (Sutton, 1988; Bertsekas and Tsitsiklis, 1996), eligibility traces (Singh and Sutton, 1996), and residual updates (Baird et al., 1995). The combination of value-based ideas with deep neural networks culminated in the Deep Q-Network (Mnih et al., 2015), which sparked a wave of extensions including Double DQN (Hasselt, 2010; Van Hasselt et al., 2016), distributional learning (Bellemare et al., 2017), and the integrative DQN-based agent (Hessel et al., 2018).

**KL-regularized RL.** KL-regularized (or entropy-regularized) reinforcement learning (RL) originated from the maximum-entropy formulation of Ziebart et al. (2008); Ziebart (2010); Neu et al. (2017), where a Kullback–Leibler (KL) penalty encourages policies to stay close to a reference distribution while optimizing reward. Different styles of algorithms have emerged from this line of work, including Soft Q-Learning (SQL) style algorithms such as SQL itself (Haarnoja et al., 2017; Schulman et al., 2017a; Guo et al., 2022), Soft Actor-Critic (SAC) style algorithms like PCL (Nachum et al., 2017), SAC (Haarnoja et al., 2018), DQO (Ji et al., 2024), and OREO (Wang et al., 2024), Point-Wise Direct Alignment Algorithms (DAA-pt) such as KTO (Ethayarajh et al., 2024), DRO (Richemond et al., 2024), and an online policy mirror descent variant (Team et al., 2025), and Pair-Wise Direct Alignment Algorithms (DAA-pair) like DPO (Rafailov et al., 2023, 2024) and IPO (Azar et al., 2024). In Appendix E, we provide a comprehensive discussion of the differences between TBRM and these related algorithms.

**RL training for LLM reasoning.** RL has played a pivotal role in the post-training of LLMs. The most prominent early example is reinforcement learning from human feedback (RLHF; Ouyang et al., 2022; Bai et al., 2022), which uses PPO to align LLMs with human preferences. A series of subsequent works introduced contrastive learning objectives based on pairwise datasets (Rafailov

et al., 2024; Zhao et al., 2023; Azar et al., 2024; Tang et al., 2024), or verification-driven objectives using a binary verifier (Ethayarajh et al., 2024; Chen et al., 2025). The release of OpenAI's O1 (Jaech et al., 2024) and DeepSeek's R1 (Guo et al., 2025) marked a new era of RL algorithms for LLMs—particularly for reasoning tasks—by framing the response generation process as a Markov Decision Process (MDP) and using rule-based verifiers to provide reward signals. Numerous studies have demonstrated and analyzed the effectiveness of RL algorithms in enhancing LLM reasoning capabilities, with PPO and GRPO emerging as the most widely adopted approaches. Prior studies, such as SimpleRL-Zoo (Zeng et al., 2025), Open Reasoner Zero (Hu et al., 2025), Light-r1 (Wen et al., 2025), Logic-rl (Xie et al., 2025), and Skywork-OR1 (He et al., 2025), fall in this category. Variants of these algorithms have been proposed to further improve performance (Zhang et al., 2025). For instance, DAPO (Yu et al., 2025) enhances GRPO with techniques like clip-higher, dynamic sampling, and token-level policy gradient loss, achieving strong results on AIME24. Dr. GRPO (Liu et al., 2025) addresses optimization bias in GRPO to improve token efficiency, while CPPO (Lin et al., 2025) reduces its computational cost by skipping rollouts with low advantages. VC-PPO (Yuan et al., 2025b) resolves PPO's challenges with value initialization bias and delayed reward signals through value pretraining and decoupled-GAE. Building on this, VAPO (Yuan et al., 2025a) improve DAPO further by incorporating selected techniques from VC-PPO. Additionally, several works explore REINFORCE (Williams and Peng, 1991; Williams, 1992) style RL algorithms, including ReMax (Li et al., 2023), REINFORCE++ (Hu, 2025), RAFT++ (Xiong et al., 2025), and RLOO (Kool et al., 2019; Ahmadian et al., 2024). However, all these approaches rely on policy-based methods or their variants. In contrast, our method adopts a value-based, off-policy RL approach that is principled, efficient, and theoretically grounded.

## B Technical Tools

We now present Lemma 3 as a soft performance difference lemma with arbitrary reference function. Lemma 3 holds generally for KL-regularized RL, which uses a slightly different form of Bellman operator defined as below,

$$(\mathcal{T}_\beta^\pi f)(s_h, a_h) \coloneqq r(s_h, a_h) + \mathbb{E}_{s_{h+1} \sim \mathcal{P}(\cdot \mid s_h, a_h), a_{h+1} \sim \pi(\cdot \mid s_{h+1})} \left[ f(s_{h+1}, a_{h+1}) - \beta \log \frac{\pi(a_{h+1} \mid s_{h+1})}{\pi_{\text{ref}}(a_{h+1} \mid s_{h+1})} \right].$$

We present this slightly different version as this lemma is more generally applicable than the results in this paper.

**Lemma 3** (Soft Performance Difference Lemma via Reference Function). *For any function $f$ as well as any policies $\pi^\dagger$ and $\pi$, we have*

$$J_\beta(\pi^\dagger) - J_\beta(\pi) = \mathbb{E}_{\pi^\dagger} \left[ \sum_{h=1}^H \left( (\mathcal{T}_\beta^\pi f)(s_h, a_h) - f(s_h, a_h) \right) \right] + \mathbb{E}_\pi \left[ \sum_{h=1}^H \left( f(s_h, a_h) - (\mathcal{T}_\beta^\pi f)(s_h, a_h) \right) \right]$$

$$+ \mathbb{E}_{\pi^\dagger} \left[ \sum_{h=1}^H \left( f(s_h, a_h) - \beta \log \frac{\pi^\dagger(a_h \mid s_h)}{\pi_{\text{ref}}(a_h \mid s_h)} - \mathbb{E}_{a_h \sim \pi(\cdot \mid s_h)} \left[ f(s_h, a_h) - \beta \log \frac{\pi(a_h \mid s_h)}{\pi_{\text{ref}}(a_h \mid s_h)} \right] \right) \right].$$

***Proof of Lemma 3.*** We first prove the soft performance difference via $Q_\beta^\pi$ as the reference,

$$J_\beta(\pi^\dagger) - J_\beta(\pi)$$

$$= \mathbb{E}_{\pi^\dagger} \left[ \sum_{h=1}^H \left( r(s_h, a_h) - \beta \log \frac{\pi^\dagger(a_h \mid s_h)}{\pi_{\text{ref}}(a_h \mid s_h)} \right) \right] - J_\beta(\pi)$$

$$= \mathbb{E}_{\pi^\dagger} \left[ \sum_{h=1}^H \left( r(s_h, a_h) - \beta \log \frac{\pi^\dagger(a_h \mid s_h)}{\pi_{\text{ref}}(a_h \mid s_h)} - V_\beta^\pi(s_h) + V_\beta^\pi(s_h) \right) \right] - J_\beta(\pi)$$

$$= \mathbb{E}_{\pi^\dagger} \left[ \sum_{h=1}^H \left( r(s_h, a_h) - \beta \log \frac{\pi^\dagger(a_h \mid s_h)}{\pi_{\text{ref}}(a_h \mid s_h)} - V_\beta^\pi(s_h) + V_\beta^\pi(s_h) \right) \right] - \mathbb{E}_{s_1 \sim \rho} \left[ V_\beta^\pi(s_1) \right]$$

$$= \mathbb{E}_{\pi^\dagger} \left[ \sum_{h=1}^H \left( r(s_h, a_h) - \beta \log \frac{\pi^\dagger(a_h \mid s_h)}{\pi_{\text{ref}}(a_h \mid s_h)} - V_\beta^\pi(s_h) + V_\beta^\pi(s_h) \right) - V_\beta^\pi(s_1) \right]$$

$$= \mathbb{E}_{\pi^\dagger} \left[ \sum_{h=1}^{H} \left( r(s_h, a_h) + \mathbb{E}_{s_{h+1} \sim \mathcal{P}(\cdot | s_h, a_h)} \left[ V_\beta^\pi(s_{h+1}) \right] - \beta \log \frac{\pi^\dagger(a_h \mid s_h)}{\pi_{\mathsf{ref}}(a_h \mid s_h)} - V_\beta^\pi(s_h) \right) \right]$$

$$= \mathbb{E}_{\pi^\dagger} \left[ \sum_{h=1}^{H} \left( Q_\beta^\pi(s_h, a_h) - \beta \log \frac{\pi^\dagger(a_h \mid s_h)}{\pi_{\mathsf{ref}}(a_h \mid s_h)} - \mathbb{E}_{a_h \sim \pi(\cdot | s_h)} \left[ Q_\beta^\pi(s_h, \pi) + \beta \log \frac{\pi(a_h \mid s_h)}{\pi_{\mathsf{ref}}(a_h \mid s_h)} \right] \right) \right].$$

Next, consider an arbitrary reference function $f$, we define the augment reward $r_f$ as

$$r_f(s_h, a_h) := f(s_h, a_h) - \mathbb{E}_{s_{h+1} \sim \mathcal{P}(\cdot | s_h, a_h), a_{h+1} \sim \pi(\cdot | s_{h+1})} \left[ f(s_{h+1}, a_{h+1}) - \beta \log \frac{\pi(a_{h+1} \mid s_{h+1})}{\pi_{\mathsf{ref}}(a_{h+1} \mid s_{h+1})} \right]$$

$$= f(s_h, a_h) - (\mathcal{T}_\beta^\pi f)(s_h, a_h) + r(s_h, a_h),$$

for any $(s_h, a_h) \in \mathcal{S}_h \times \mathcal{A}_h$. This means $f$ is the fixed point of $\mathcal{T}_\beta^\pi$ with replacing reward function $r$ by $r_f$. We use $Q_{\beta, r_f}^\pi$ to denote the soft Q-function with replacing reward function $r$ by $r_f$, and we immediately have $f \equiv Q_{\beta, r_f}^\pi$. We also use $J_{\beta, r_f}(\pi)$ to denote the soft return of policy $\pi$ with replacing reward function $r$ by $r_f$. Then

$$J_\beta(\pi^\dagger) - J_\beta(\pi)$$
$$= J_\beta(\pi^\dagger) - J_{\beta, r_f}(\pi^\dagger) + J_{\beta, r_f}(\pi^\dagger) - J_{\beta, r_f}(\pi) + J_{\beta, r_f}(\pi) - J_\beta(\pi)$$
$$= \mathbb{E}_{\pi^\dagger} \left[ \sum_{h=1}^{H} (r(s_h, a_h) - r_f(s_h, a_h)) \right] + \mathbb{E}_\pi \left[ \sum_{h=1}^{H} (r_f(s_h, a_h) - r(s_h, a_h)) \right]$$
$$+ \mathbb{E}_{\pi^\dagger} \left[ \sum_{h=1}^{H} \left( f(s_h, a_h) - \beta \log \frac{\pi^\dagger(a_h \mid s_h)}{\pi_{\mathsf{ref}}(a_h \mid s_h)} - \mathbb{E}_{a_h \sim \pi(\cdot | s_h)} \left[ f(s_h, a_h) - \beta \log \frac{\pi(a_h \mid s_h)}{\pi_{\mathsf{ref}}(a_h \mid s_h)} \right] \right) \right]$$
$$\text{(by the soft performance difference via reference as } f \equiv Q_{\beta, r_f}^\pi)$$
$$= \mathbb{E}_{\pi^\dagger} \left[ \sum_{h=1}^{H} ((\mathcal{T}_\beta^\pi f)(s_h, a_h) - f(s_h, a_h)) \right] + \sum_{h=1}^{H} \mathbb{E}_{d_h^\pi} \left[ f(s_h, a_h) - (\mathcal{T}_\beta^\pi f)(s_h, a_h) \right]$$
$$+ \mathbb{E}_{\pi^\dagger} \left[ \sum_{h=1}^{H} \left( f(s_h, a_h) - \beta \log \frac{\pi^\dagger(a_h \mid s_h)}{\pi_{\mathsf{ref}}(a_h \mid s_h)} - \mathbb{E}_{a_h \sim \pi(\cdot | s_h)} \left[ f(s_h, a_h) - \beta \log \frac{\pi(a_h \mid s_h)}{\pi_{\mathsf{ref}}(a_h \mid s_h)} \right] \right) \right].$$

This completes the proof. □

When specializing to the Bellman operator in this paper as in Section 2.3, Lemma 3 becomes

$$\frac{J_\beta(\pi^\dagger) - J_\beta(\pi)}{\beta} = \mathbb{E}_{\pi^\dagger} \left[ \sum_{h=1}^{H} ((\mathcal{T}_\beta^\pi Q)(s_h, a_h) - Q(s_h, a_h)) \right] + \mathbb{E}_\pi \left[ \sum_{h=1}^{H} (Q(s_h, a_h) - (\mathcal{T}_\beta^\pi Q)(s_h, a_h)) \right]$$
$$+ \mathbb{E}_{\pi^\dagger} \left[ \sum_{h=1}^{H} \left( Q(s_h, a_h) - \log \pi^\dagger(a_h \mid s_h) - \mathbb{E}_{a_h \sim \pi(\cdot | s_h)} [Q(s_h, a_h) - \log \pi(a_h \mid s_h)] \right) \right].$$

As corollary, we have the following upper bound on the sub-optimality of any policy induced by a value function.

**Corollary 4.** *Suppose that the policy $\widehat{\pi} = \pi_Q$ is induced by a value function $Q$, i.e.,*

$$\widehat{\pi}(a \mid s) = \frac{\exp(Q(s, a))}{\sum_{a' \in \mathcal{A}} \exp(Q(s, a'))}.$$

*Then it holds that*

$$J_\beta(\pi^\star) - J_\beta(\widehat{\pi}) \leq 2\beta \max_{\pi \in \{\pi^\star, \widehat{\pi}\}} \left| \mathbb{E}_\pi \left[ \sum_{h=1}^{H} (Q(s_h, a_h) - (\mathcal{T}_\beta Q)(s_h, a_h)) \right] \right|$$

***Proof of Corollary 4.*** By Lemma 3, we have
$$J_\beta(\pi^\star) - J_\beta(\widehat{\pi})$$

$$= \beta \mathbb{E}_{\pi^\star} \left[ \sum_{h=1}^{H} \left( (\mathcal{T}_\beta^{\widehat{\pi}} Q)(s_h, a_h) - Q(s_h, a_h) \right) \right] + \beta \mathbb{E}_{\widehat{\pi}} \left[ \sum_{h=1}^{H} \left( Q(s_h, a_h) - (\mathcal{T}_\beta^{\widehat{\pi}} Q)(s_h, a_h) \right) \right]$$

$$+ \beta \mathbb{E}_{\pi^\star} \left[ \sum_{h=1}^{H} \left( Q(s_h, a_h) - \log \pi^\star(a_h \mid s_h) - \mathbb{E}_{a_h' \sim \widehat{\pi}(\cdot \mid s_h)} [Q(s_h, a_h') - \log \widehat{\pi}(a_h' \mid s_h)] \right) \right]$$

$$\leq \beta \mathbb{E}_{\pi^\star} \left[ \sum_{h=1}^{H} \left( (\mathcal{T}_\beta Q)(s_h, a_h) - Q(s_h, a_h) \right) \right] + \beta \mathbb{E}_{\widehat{\pi}} \left[ \sum_{h=1}^{H} \left( Q(s_h, a_h) - (\mathcal{T}_\beta Q)(s_h, a_h) \right) \right]$$

$$\leq 2\beta \max_{\pi} \left| \mathbb{E}_{\pi} \left[ \sum_{h=1}^{H} \left( Q(s_h, a_h) - (\mathcal{T}_\beta Q)(s_h, a_h) \right) \right] \right|,$$

where the first inequality follows from the definition that

$$\widehat{\pi}(\cdot \mid s_h) = \operatorname*{argmax}_{p \in \Delta(\mathcal{A})} \mathbb{E}_{a_h' \sim p} [Q(s_h, a_h') - \log p(a_h')], \quad \forall s_h \in \mathcal{S}_h,$$

and hence

$$\mathbb{E}_{a_h \sim \pi^\star(\cdot \mid s_h)} [Q(s_h, a_h) - \log \pi^\star(a_h \mid s_h)] \leq \mathbb{E}_{a_h' \sim \widehat{\pi}(\cdot \mid s_h)} [Q(s_h, a_h') - \log \widehat{\pi}(a_h' \mid s_h)].$$

$\square$

## C  Change of Trajectory Measure with Concentrability

We first state the following lemma, which follows immediately from the definition of $\chi^2$-divergence, see e.g. (Polyanskiy and Wu, 2025, Eq. (7.91))

**Lemma 5.** *For distribution $P, Q \in \Delta(\mathcal{X})$ and function $F : \mathcal{X} \to \mathbb{R}$ such that $\mathbb{E}_Q[F(X)] = 0$, it holds that*

$$(\mathbb{E}_P F(X))^2 \leq \chi^2 (P \parallel Q) \cdot \mathbb{E}_Q F(X)^2.$$

***Proof of Lemma 1.*** We only need to prove that for any function $f : \mathcal{S} \times \mathcal{A} \to \mathbb{R}$,

$$\left( \mathbb{E}_\pi \left[ \sum_{h=1}^{H} f(s_h, a_h) \right] \right)^2 \leq \left( 1 + \sum_{h=1}^{H} \chi^2 (d_h^\pi \parallel d_h^{\pi_{\text{ref}}}) \right) \cdot \mathbb{E}_{\pi_{\text{ref}}} \left[ \left( \sum_{h=1}^{H} f(s_h, a_h) \right)^2 \right]. \quad (7)$$

First of all, we construct function $\bar{f} : \mathcal{S} \to \mathbb{R}$ as

$$\bar{f}(s_h) := \mathbb{E}_\mu \left[ \sum_{h'=h}^{H} f(s_{h'}, a_{h'}) \;\middle|\; s_h \right], \qquad \forall h \in [H], s_h \in \mathcal{S}_h. \quad (8)$$

Then, it is direct to verify that for $h \in [H]$, $s_h \in \mathcal{S}_h$,

$$\bar{f}(s_h) := \mathbb{E}_\mu \left[ f(s_h, a_h) + \bar{f}(s_{h+1}) \;\middle|\; s_h \right],$$

where we adopt the notation that $s_{H+1}$ is a deterministic terminal state and $\bar{f}(s_{H+1}) = 0$. Then, we expand

$$\mathbb{E}_\mu \left[ \left( \sum_{h=1}^{H} f(s_h, a_h) \right)^2 \right]$$

$$= \mathbb{E}_\mu \left[ \left( \bar{f}(s_1) + \sum_{h=1}^{H} \left[ f(s_h, a_h) + \bar{f}(s_{h+1}) - \bar{f}(s_h) \right] \right)^2 \right]$$

$$= \mathbb{E}_\mu \left[ \bar{f}(s_1)^2 + \sum_{h=1}^{H} \left( f(s_h, a_h) + \bar{f}(s_{h+1}) - \bar{f}(s_h) \right)^2 \right]$$

$$+ \mathbb{E}_\mu \left[ \sum_{1 \le h \le H} \bar{f}(s_1) \left( f(s_h, a_h) + \bar{f}(s_{h+1}) - \bar{f}(s_h) \right) \right]$$

$$+ \mathbb{E}_\mu \left[ \sum_{1 \le h' < h \le H} \left( f(s_{h'}, a_{h'}) + \bar{f}(s_{h'+1}) - \bar{f}(s_{h'}) \right) \left( f(s_h, a_h) + \bar{f}(s_{h+1}) - \bar{f}(s_h) \right) \right].$$

Therefore, using the Markov property, it holds that for any $h \in [H]$,

$$\mathbb{E}_\mu \left[ f(s_h, a_h) + \bar{f}(s_{h+1}) - \bar{f}(s_h) | s_1, a_1, \cdots, s_h \right] = \mathbb{E}_\mu \left[ f(s_h, a_h) + \bar{f}(s_{h+1}) \mid s_h \right] - \bar{f}(s_h) = 0,$$

and hence we can deduce that

$$\mathbb{E}_\mu \left[ \left( \sum_{h=1}^H f(s_h, a_h) \right)^2 \right] = \mathbb{E}_\mu \left[ \bar{f}(s_1)^2 + \sum_{h=1}^H \left( f(s_h, a_h) + \bar{f}(s_{h+1}) - \bar{f}(s_h) \right)^2 \right]. \quad (9)$$

Next, for every $h \in [H]$, we apply Lemma 5 on the function $(s_h, a_h, s_{h+1}) \mapsto f(s_h, a_h) - \bar{f}(s_h) + \bar{f}(s_{h+1})$ to derive

$$\begin{aligned} & \left( \mathbb{E}_\pi \left[ f(s_h, a_h) - \bar{f}(s_h) + \bar{f}(s_{h+1}) \right] \right)^2 \\ & \le \chi^2 \left( d_h^\pi \parallel d_h^\mu \right) \cdot \mathbb{E}_\mu \left[ \left( f(s_h, a_h) - \bar{f}(s_h) + \bar{f}(s_{h+1}) \right)^2 \right], \end{aligned} \quad (10)$$

where we again use the fact that $\mathbb{E}_\mu \left[ f(s_h, a_h) + \bar{f}(s_{h+1}) - \bar{f}(s_h) \right] = 0$.

Furthermore, we note that $\mathbb{E}_\pi \bar{f}(s_1) = \mathbb{E}_{s_1 \sim \rho} \bar{f}(s_1) = \mathbb{E}_\mu \bar{f}(s_1)$, and hence

$$\left( \mathbb{E}_\pi \bar{f}(s_1) \right)^2 = \left( \mathbb{E}_\mu \bar{f}(s_1) \right)^2 \le \mathbb{E}_\mu \bar{f}(s_1)^2. \quad (11)$$

Therefore, combining the inequalities above, we have

$$\left( \mathbb{E}_\pi \left[ \sum_{h=1}^H f(s_h, a_h) \right] \right)^2$$

$$= \left( \bar{f}(s_1) + \sum_{h=1}^H \mathbb{E}_\pi \left[ f(s_h, a_h) - \bar{f}(s_h) + \bar{f}(s_{h+1}) \right] \right)^2$$

$$\overset{(i)}{\le} \left( \sqrt{\mathbb{E}_\mu \bar{f}(s_1)^2} + \sum_{h=1}^H \sqrt{\chi^2 \left( d_h^\pi \parallel d_h^\mu \right) \cdot \mathbb{E}_\mu \left[ \left( f(s_h, a_h) - \bar{f}(s_h) + \bar{f}(s_{h+1}) \right)^2 \right]} \right)^2$$

$$\overset{(ii)}{\le} \left( 1 + \sum_{h=1}^H \chi^2 \left( d_h^\pi \parallel d_h^\mu \right) \right) \left( \mathbb{E}_\mu \bar{f}(s_1)^2 + \sum_{h=1}^H \mathbb{E}_\mu \left( f(s_h, a_h) - \bar{f}(s_h) + \bar{f}(s_{h+1}) \right)^2 \right)$$

$$\overset{(iii)}{=} \left( 1 + \sum_{h=1}^H \chi^2 \left( d_h^\pi \parallel d_h^\mu \right) \right) \cdot \mathbb{E}_\mu \left[ \left( \sum_{h=1}^H f(s_h, a_h) \right)^2 \right],$$

where $(i)$ uses Eq. (10) and Eq. (11), $(ii)$ uses Cauchy-Schwarz inequality, and (iii) uses Eq. (9). Hence Eq. (7) is verified. □

Recall the definition of state-action concentrability (e.g. Eq. (5) in Jia et al. (2025)):

$$C_{\mathtt{sa}}(\pi; \mu) = \max_{h \in [H]} \sup_{s_h \in \mathcal{S}_h, a_h \in \mathcal{A}} \frac{d_h^\pi(s_h, a_h)}{d_h^\mu(s_h, a_h)}.$$

We have following direct corollary of Lemma 1, after noticing that the $\chi^2$ divergence can be upper bounded by the state concentrability. However, we remark that the state concentrability might yield a much more pessimistic bound compared to the upper bound of Lemma 1.

**Corollary 6.** *Given MDP $M = (H, \mathcal{S}, \mathcal{A}, \mathcal{P}, r, \rho)$. We use $d_h^\pi(\cdot), d_h^\mu(\cdot) \in \Delta(\mathcal{S}_h \times \mathcal{A})$ to denote the occupancy measure of the MDP under policy $\pi$ and $\mu$. Then we have*

$$\sup_f \frac{\left(\mathbb{E}_\pi \left[\sum_{h=1}^H f(s_h, a_h)\right]\right)^2}{\mathbb{E}_\mu \left[\left(\sum_{h=1}^H f(s_h, a_h)\right)^2\right]} \leq H \cdot C_{\mathsf{sa}}(\pi; \mu),$$

*where the supremum is over all functions $f : \mathcal{S} \times \mathcal{A} \to \mathbb{R}$.*

***Proof of Corollary 6.*** In view of Lemma 1, we only need to verify that

$$1 + \chi^2 \left(d_h^\pi \parallel d_h^\mu\right) \leq C_{\mathsf{sa}}. \tag{12}$$

We have

$$1 + \chi^2 \left(d_h^\pi \parallel d_h^\mu\right) = \sum_{s_h \in \mathcal{S}_h, a_h \in \mathcal{A}} \frac{d_h^\pi(s_h, a_h)^2}{d_h^\mu(s_h, a_h)}$$

$$\leq \sum_{s_h \in \mathcal{S}_h, a_h \in \mathcal{A}} d_h^\pi(s_h, a_h) \cdot C_{\mathsf{sa}}(\pi; \mu) = C_{\mathsf{sa}}(\pi; \mu),$$

and Eq. (12) is verified. $\qquad\square$

# D   Proof of Theorem 2

Recall that the TBRM loss defined in Eq. (4) is given by

$$\mathcal{L}_{\mathcal{D}}^{\mathsf{TBRM}}(\theta) := \frac{1}{|\mathcal{D}|} \sum_{\tau \in \mathcal{D}} \left( \mathsf{logit}_\theta(s_1, a_1) - \frac{r(\tau)}{\beta} - \log \pi_{\mathsf{ref}}(\tau) + \sum_{h=2}^H \log \pi_\theta(a_h \mid s_h) \right)^2. \tag{13}$$

In the following, to simplify presentation, we define

$$f_{\theta,1}(s_1, a_1) = \mathsf{logit}_\theta(s_1, a_1) - \frac{r(s_1, a_1)}{\beta} - \log \pi_{\mathsf{ref}}(a_1 \mid s_1),$$

$$f_{\theta,h}(s_h, a_h) = \log \pi_\theta(a_h \mid s_h) - \frac{r(s_h, a_h)}{\beta} - \log \pi_{\mathsf{ref}}(a_h \mid s_h), \qquad \forall h > 1.$$

and

$$f_\theta(\tau) := \mathsf{logit}_\theta(s_1, a_1) - \frac{r(\tau)}{\beta} - \log \pi_{\mathsf{ref}}(\tau) + \sum_{h=2}^H \log \pi_\theta(a_h \mid s_h)$$

$$= \sum_{h=1}^H f_{\theta,h}(s_h, a_h), \qquad \forall \tau = (s_1, a_1, \cdots, s_H, a_H).$$

**Uniform convergence.** Before applying Lemma 1, we need to first relate the empirical loss $\mathcal{L}_{\mathcal{D}}^{\mathsf{TBRM}}$ to the population loss. We introduce the following assumption on the parametrization.

**Assumption 2** (Bounded and smooth parametrization). *There exists constant $C_\Theta \geq 1$ and parameter $L_\Theta$ such that for any $\theta \in \Theta$, it holds that $\forall s \in \mathcal{S}, a \in \mathcal{A}$,*

$$|\mathsf{logit}_\theta(s, a)| \leq \frac{C_\Theta}{\beta}, \qquad \|\nabla \mathsf{logit}_\theta(s, a)\|_* \leq L_\Theta,$$

*where $\|\cdot\|_*$ is the dual norm of $\|\cdot\|$. We also assume $|\log \pi_{\mathsf{ref}}(a \mid s)| \leq \frac{C_\Theta}{\beta} \, \forall s \in \mathcal{S}, a \in \mathcal{A}$.*

**Lemma 7.** *Fix $\delta \in (0, 1)$. Suppose that Assumption 2 holds. Then with probability at least $1 - \delta$ (over the randomness of the dataset $\mathcal{D}$),*

$$\mathbb{E}_{\tau \sim \mu} f_\theta(\tau)^2 \leq 2\mathcal{L}_{\mathcal{D}}^{\mathsf{TBRM}}(\theta) + \frac{\varepsilon_{\mathsf{stat}}(N)}{\beta^2},$$

*where the statistical error $\varepsilon_{\mathsf{stat}}(N)$ is defined as*

$$\varepsilon_{\mathsf{stat}}(N) := c \cdot C_\Theta^2 H^2 \left( \frac{\log(\mathcal{N}(\Theta, \alpha)/\delta)}{N} + HL_\Theta \beta \alpha \right)$$

*where $c > 0$ is a large absolute constant and $\alpha \geq 0$ is a fixed parameter.*

The above upper bound can be further simplified by the standard assumption on the covering number of $\Theta$.

**Assumption 3** (Parametric function class). *The parameter space $\Theta \subseteq \{\theta \in \mathbb{R}^d : \|\theta\| \leq R\}$. In this case, we write $\dim(\Theta) = d$.*

Under Assumption 3, it is clear that $\log \mathcal{N}(\Theta, \alpha) \leq O(d \log(1/\alpha))$ for all $\alpha > 0$ (see e.g., Wainwright (2019)). Therefore, Lemma 7 implies that $\varepsilon_{\mathsf{stat}}(N) \asymp \frac{H^2}{N}$ (up to poly-logarithmic factors).

**Bounding the sub-optimality.** Under Assumption 1, it is clear that $f_{\theta^\star} \equiv 0$ and hence $\mathcal{L}_{\mathcal{D}}^{\mathsf{TBRM}}(\theta^\star) = 0$ (as we have argued in Section 3.1). Therefore, using the condition that $\mathcal{L}_{\mathcal{D}}^{\mathsf{TBRM}}(\widehat{\theta}) - \inf_{\theta \in \Theta} \mathcal{L}_{\mathcal{D}}^{\mathsf{TBRM}}(\theta) \leq \varepsilon_{\mathsf{opt}}$, we have

$$\mathbb{E}_{\tau \sim \mu} f_{\widehat{\theta}}(\tau)^2 \leq 2\varepsilon_{\mathsf{opt}} + \beta^{-2}\varepsilon_{\mathsf{stat}}(N).$$

In the following, we denote $\widehat{\pi} := \pi_{\widehat{\theta}}$, $\widehat{Q}(s,a) := \mathsf{logit}_{\widehat{\theta}}(s,a)$, and $\widehat{V}$ be the corresponding value function. Then, by Corollary 4, we have

$$J_\beta(\pi^\star) - J_\beta(\widehat{\pi}) \leq 2\beta \max_{\pi \in \{\pi^\star, \widehat{\pi}\}} \left| \mathbb{E}_\pi \left[ \sum_{h=1}^{H} \left( \widehat{Q}(s_h, a_h) - \left( \mathcal{T}_\beta \widehat{Q} \right)(s_h, a_h) \right) \right] \right|.$$

Note that the MDP is deterministic, and hence $\left( \mathcal{T}_\beta \widehat{Q} \right)(s_h, a_h) = R_\beta(s_h, a_h) + \widehat{V}(s_{h+1})$ holds deterministically. Therefore, for any fixed policy $\pi$, we have

$$\mathbb{E}_\pi \left[ \sum_{h=1}^{H} \left( \widehat{Q}(s_h, a_h) - \left( \mathcal{T}_\beta \widehat{Q} \right)(s_h, a_h) \right) \right]$$

$$= \mathbb{E}_\pi \left[ \sum_{h=1}^{H} \left( \widehat{Q}(s_h, a_h) - R_\beta(s_h, a_h) - \widehat{V}(s_{h+1}) \right) \right]$$

$$= \mathbb{E}_\pi \left[ \widehat{Q}(s_1, a_1) - \sum_{h=1}^{H} R_\beta(s_h, a_h) + \sum_{h=1}^{H} \left[ \widehat{Q}(s_h, a_h) - \widehat{V}(s_h) \right] \right]$$

$$= \mathbb{E}_\pi \left[ \widehat{Q}(s_1, a_1) - R_\beta(\tau) + \sum_{h=1}^{H} \log \pi_{\widehat{\theta}}(a_h \mid s_h) \right] = \mathbb{E}_\pi \left[ f_{\widehat{\theta}}(\tau) \right].$$

Further, by Lemma 1, it holds that for any $\pi = \pi_\theta$,

$$\left( \mathbb{E}_\pi \left[ f_{\widehat{\theta}}(\tau) \right] \right)^2 \leq H C_{\mathsf{conc}}(\mu) \cdot \mathbb{E}_{\tau \sim \mu} f_{\widehat{\theta}}(\tau)^2 \leq H C_{\mathsf{conc}}(\mu) \left( 2\varepsilon_{\mathsf{opt}} + \beta^{-2}\varepsilon_{\mathsf{stat}}(N) \right).$$

Therefore, we can conclude that

$$J_\beta(\pi^\star) - J_\beta(\widehat{\pi}) \leq 2\beta \max_{\pi \in \{\pi^\star, \widehat{\pi}\}} \sqrt{\mathbb{E}_\pi \left[ f_{\widehat{\theta}}(\tau) \right]} \leq 2\sqrt{H C_{\mathsf{conc}}(\mu)(2\beta^2 \varepsilon_{\mathsf{opt}} + \varepsilon_{\mathsf{stat}}(N))}.$$

This is the desired upper bound. $\qquad\qquad\qquad\qquad\qquad\qquad\qquad\qquad\qquad\qquad\qquad\quad\square$

### D.1 Proof of Lemma 7

By Assumption 2, it holds that $|f_\theta(\tau)| \leq B := \frac{2C_\Theta + 1}{\beta}$ for any $\theta \in \Theta$ and any trajectory $\tau$. Using Freedman's inequality with the standard union bound, we have the following: with probability at least $1 - \delta$ (over the randomness of the dataset $\mathcal{D}$), for all $\theta \in \Theta$,

$$\mathbb{E}_{\tau \sim \mu} f_\theta(\tau)^2 \leq 2\mathcal{L}_{\mathcal{D}}^{\mathsf{TBRM}}(\theta) + c_0 B^2 \left( \frac{\log \mathcal{N}(\mathcal{F}, \alpha) + \log(1/\delta)}{N} + B^{-1}\alpha \right),$$

where $\mathcal{F} = \{f_\theta : \theta \in \Theta\}$ is the function class induced by $\Theta$, and $c_0 > 0$ is an absolute constant.

Next, for any fixed trajectory $\tau$, it holds that

$$\nabla_\theta f_\theta(\tau) = \nabla_\theta \mathsf{logit}_\theta(s_1, a_1) + \sum_{h=2}^{H} \nabla_\theta \log \pi_\theta(a_h \mid s_h)$$

$$= \nabla_\theta \mathsf{logit}_\theta(s_1, a_1) + \sum_{h=2}^{H} \left[ \nabla_\theta \mathsf{logit}_\theta(s_h, a_h) - \mathbb{E}_{a_h' \sim \pi_\theta(\cdot | s_h)} \nabla_\theta \mathsf{logit}_\theta(s_h, a_h') \right].$$

Hence, we can upper bound $\|\nabla_\theta f_\theta(\tau)\|_* \leq 2HL_\Theta$. This immediately implies that

$$\|f_\theta - f_{\theta'}\|_\infty = \sup_\tau |f_\theta(\tau) - f_{\theta'}(\tau)| \leq 2HL_\Theta \|\theta - \theta'\|, \qquad \forall \theta, \theta' \in \Theta.$$

Therefore, we have

$$\mathcal{N}(\mathcal{F}, \alpha) \leq \mathcal{N}\left( \Theta, \frac{\alpha}{2HL_\Theta} \right).$$

Combining the inequalities above and rescaling $\alpha \leftarrow 2HL_\Theta \alpha$ completes the proof. $\qquad \square$

# E   Comparison with Related Algorithms

In this section, we compare TBRM with other related algorithms in detail. We group the related algorithms into the following categories:

- **Soft Q-Learning (SQL) Style**: SQL (Haarnoja et al., 2017; Schulman et al., 2017a; Guo et al., 2022)
- **Soft Actor-Critic (SAC) Style**: PCL (Nachum et al., 2017), SAC (Haarnoja et al., 2018), DQO (Ji et al., 2024), OREO (Wang et al., 2024)
- **Point-Wise Direct Alignment Algorithms (DAA-pt)**: KTO (Ethayarajh et al., 2024), DRO (Richemond et al., 2024), online policy mirror decent variant (Team et al., 2025)
- **Pair-Wise Direct Alignment Algorithms (DAA-pair)**: DPO (Rafailov et al., 2023, 2024), IPO (Azar et al., 2024)

It is important to note that the present paper primarily addresses LLM reasoning in environments where the state space is tokenized and the base model operates autoregressively. Several algorithms mentioned above were initially developed for continuous control domains such as robotics; however, our analysis considers only their adaptation to the discrete, tokenized setting relevant to language models, as in this paper. Given space constraints, we restrict our discussion to the fundamental principles underlying each algorithmic category rather than providing exhaustive implementation details.

We first present Table 2 to summarize the key differences between TBRM and other algorithms. Note that this comparison is only for algorithm design; the consequences of these differences for theoretical guarantees are likely to be more significant. However, given that TBRM is the only algorithm here with established finite-sample guarantees under the more general MDP setting, we will leave the theoretical comparison to future work.

| Algorithms | Optimization | Single Rollout | Single Model Training | Traj. Reward Allowed |
|---|---|---|---|---|
| SQL | ✗ | ✓ | ✓ | ✗ |
| SAC | ✓ | ✓ | ✗ | ✗ |
| DAA-pt | ✓ | ✗ | ✓ | ✓ |
| DAA-pair | ✓ | ✗ | ✓ | ✓ |
| TBRM | ✓ | ✓ | ✓ | ✓ |

Table 2: Comparison between TBRM and related algorithms in terms of algorithm design.

For the ease of comparison, we rewrite the loss function of TBRM as follows, by the definition of the autoregressive function approximation:

$$\mathcal{L}_\mathcal{D}^{\mathsf{TBRM}}(\theta) = \frac{1}{|\mathcal{D}|} \sum_{\tau \in \mathcal{D}} \left( \mathsf{logit}_\theta(s_1, a_1) - \frac{r(\tau)}{\beta} - \log \pi_{\mathsf{ref}}(\tau) + \sum_{h=2}^{H} \log \pi_\theta(a_h \mid s_h) \right)^2$$

$$= \frac{1}{|\mathcal{D}|} \sum_{\tau \in \mathcal{D}} \left( V_\theta(s_1) - \frac{r(\tau)}{\beta} + \sum_{h=1}^{H} \log \frac{\pi_\theta(a_h \mid s_h)}{\pi_{\mathsf{ref}}(a_h \mid s_h)} \right)^2.$$

**Comparing TBRM with Soft Q-Learning Style Algorithms.** The soft Q-learning based algorithms are typically iterative algorithms, formulated as two different versions: single-step case and multi-step case. We consider the loss used in Guo et al. (2022) which is motivated by *path consistency learning* (PCL; Nachum et al., 2017). The single-step case is then formulated as

$$\theta_{t+1} \leftarrow \arg\min_\theta \mathcal{L}_\mathcal{D}^{\mathsf{SQL\text{-}s}}(\theta; \theta_t) \coloneqq \frac{1}{|\mathcal{D}|} \sum_{\tau \in \mathcal{D}} \sum_{h=1}^H \left( V_{\theta_t}(s_h) - \frac{r(s_h, a_h)}{\beta} + \log \frac{\pi_\theta(a_h \mid s_h)}{\pi_{\mathsf{ref}}(a_h \mid s_h)} - V_{\theta_t}(s_{h+1}) \right)^2,$$

while the multi-step case is formulated as

$$\theta_{t+1} \leftarrow \arg\min_\theta \mathcal{L}_\mathcal{D}^{\mathsf{SQL\text{-}m}}(\theta; \theta_t) \coloneqq \frac{1}{|\mathcal{D}|} \sum_{\tau \in \mathcal{D}} \sum_{h=1}^H \left( V_{\theta_t}(s_h) - \sum_{h'=h}^H \frac{r(s_{h'}, a_{h'})}{\beta} + \sum_{h'=h}^H \log \frac{\pi_\theta(a_{h'} \mid s_{h'})}{\pi_{\mathsf{ref}}(a_{h'} \mid s_{h'})} \right)^2.$$

If we want to exactly match the original loss of soft Q-learning (Haarnoja et al., 2017; Schulman et al., 2017a), then these should be rewritten as

$$\theta_{t+1} \leftarrow \arg\min_\theta \widetilde{\mathcal{L}}_\mathcal{D}^{\mathsf{SQL\text{-}s}}(\theta; \theta_t) \coloneqq \frac{1}{|\mathcal{D}|} \sum_{\tau \in \mathcal{D}} \sum_{h=1}^H \left( \mathsf{logit}_\theta(s_h) - \frac{r(s_h, a_h)}{\beta} - \log \pi_{\mathsf{ref}}(a_h \mid s_h) - V_{\theta_t}(s_{h+1}) \right)^2,$$

$$\theta_{t+1} \leftarrow \arg\min_\theta \widetilde{\mathcal{L}}_\mathcal{D}^{\mathsf{SQL\text{-}m}}(\theta; \theta_t) \coloneqq \frac{1}{|\mathcal{D}|} \sum_{\tau \in \mathcal{D}} \sum_{h=1}^H \left( \mathsf{logit}_\theta(s_h) - \sum_{h'=h}^H \frac{r(s_{h'}, a_{h'})}{\beta} - \log \pi_{\mathsf{ref}}(a_h \mid s_h) + \sum_{h'=h+1}^H \log \frac{\pi_{\theta_t}(a_{h'} \mid s_{h'})}{\pi_{\mathsf{ref}}(a_{h'} \mid s_{h'})} \right)^2.$$

There can be a more general multi-step version, which blends the multi-step return in $\mathcal{L}_\mathcal{D}^{\mathsf{SQL\text{-}m}}$ and the value bootstrap in $\mathcal{L}_\mathcal{D}^{\mathsf{SQL\text{-}s}}$, but we omit it here for brevity as our existing argument would directly extend to this case.

From the derivation above, we can identify two key distinctions between TBRM and soft Q-learning based algorithms: 1) TBRM employs direct optimization rather than an iterative approach, and 2) TBRM's loss function operates on complete trajectories rather than summing losses over individual timesteps within trajectories, hence eliminating the need for per-step reward.

**Comparing TBRM with Soft Actor-Critic Style Algorithms.** The soft actor-critic style algorithms for LLMs are similar to SQL, but they 1) introduce a separate $V$ model; 2) operate as optimization rather than iteration. In particular, in the single-step case,

$$\arg\min_{\theta, \phi} \mathcal{L}_\mathcal{D}^{\mathsf{SAC\text{-}s}}(\theta, \phi) \coloneqq \frac{1}{|\mathcal{D}|} \sum_{\tau \in \mathcal{D}} \sum_{h=1}^H \left( V_\phi(s_h) - \frac{r(s_h, a_h)}{\beta} + \log \frac{\pi_\theta(a_h \mid s_h)}{\pi_{\mathsf{ref}}(a_h \mid s_h)} - V_\phi(s_{h+1}) \right)^2,$$

while the multi-step case is

$$\arg\min_{\theta, \phi} \mathcal{L}_\mathcal{D}^{\mathsf{SAC\text{-}m}}(\theta, \phi) \coloneqq \frac{1}{|\mathcal{D}|} \sum_{\tau \in \mathcal{D}} \sum_{h=1}^H \left( V_\phi(s_h) - \sum_{h'=h}^H \frac{r(s_{h'}, a_{h'})}{\beta} + \sum_{h'=h}^H \log \frac{\pi_\theta(a_{h'} \mid s_{h'})}{\pi_{\mathsf{ref}}(a_{h'} \mid s_{h'})} \right)^2.$$

**Comparing TBRM with Point-Wise Direct Alignment Algorithms.** Perhaps surprisingly, among all four categories of algorithms, the point-wise direct alignment algorithms appear to be the most similar to TBRM, although they are derived from a different perspective (mostly from bandit formulation). We view the core objective of these algorithms as optimizing the following loss

$$\arg\min_\theta \mathcal{L}_\mathcal{D}^{\mathsf{DAA\text{-}pt}}(\theta) = \frac{1}{|\mathcal{D}|} \sum_{\tau \in \mathcal{D}} \left( \widehat{V}_\beta^\star(s_1) - \frac{r(\tau)}{\beta} + \sum_{h=1}^H \log \frac{\pi_\theta(a_h \mid s_h)}{\pi_{\mathsf{ref}}(a_h \mid s_h)} \right)^2,$$

where $\widehat{V}_\beta^\star$ is an estimate of $V_\beta^\star$ (the optimal value function for KL-regularized MDP). Note that $V_\beta^\star$ is exactly the same as the partition function in the bandit formulation. One popular way to estimate $\widehat{V}_\beta^\star$ is by using the softmax of returns from multiple rollouts for each question,[4] for example (Team et al., 2025),

$$\widehat{V}_\beta^\star(s_1) \leftarrow \beta \log \sum_{\tau \sim \pi_\theta \mid s_1} \exp\left( \frac{r(\tau)}{\beta} \right).$$

---

[4]$\widehat{V}_\beta^\star$ can also be estimated using a separate model (Richemond et al., 2024), similar to our discussion of SAC-style algorithms.

However, it is unclear whether this estimate is accurate enough, particularly when the rollout policy $\pi_\theta$ differs significantly from the optimal policy $\pi_\beta^\star$. In contrast, TBRM leverages 1) Bellman equation in KL-regularized RL (see, e.g., Section 2.3) and 2) recent advances in change of trajectory measure, which allows us to directly use $V_\theta$ instead of requiring $\widehat{V}_\beta^\star$. This approach enables TBRM to provably converge to a near-optimal policy using only a single rollout per prompt, while maintaining the advantages of a direct optimization algorithm.

**Comparing TBRM with Pair-Wise Direct Alignment Algorithms.**    Under the perspective above, we can view the motivation of pair-wise direct alignment algorithms as using a pair of responses from the same question $s_1$ to cancel the need for $V_\theta(s_1)$, which leads to the following objective:

$$\underset{\theta}{\operatorname{argmin}} \, \mathcal{L}_\mathcal{D}^{\mathsf{DAA\text{-}pair}}(\theta) = \frac{1}{|\mathcal{D}|} \sum_{(\tau, \tau') \in \mathcal{D}} \left( \sum_{h=1}^H \log \frac{\pi_\theta(a_h \mid s_h)}{\pi_{\mathsf{ref}}(a_h \mid s_h)} - \sum_{h=1}^H \log \frac{\pi_\theta(a_h' \mid s_h')}{\pi_{\mathsf{ref}}(a_h' \mid s_h')} - \frac{r(\tau)}{\beta} + \frac{r(\tau')}{\beta} \right)^2 .$$

Comparing with TBRM, the pair-wise direct alignment algorithms are basically optimizing the difference between the Bellman residuals of two trajectories.

# F    Hard Instances for Iterative Algorithms

In this section, we demonstrate the advantages of direct optimization (TBRM) over its iterative variant using a simple but illustrative hard instance.

By the autoregressive function approximation definition, we can rewrite the loss function of TBRM as follows:

$$
\begin{aligned}
\mathcal{L}_\mathcal{D}^{\mathsf{TBRM}}(\theta) &= \frac{1}{|\mathcal{D}|} \sum_{\tau \in \mathcal{D}} \left( \mathsf{logit}_\theta(s_1, a_1) - \frac{r(\tau)}{\beta} - \log \pi_{\mathsf{ref}}(\tau) + \sum_{h=2}^H \log \pi_\theta(a_h \mid s_h) \right)^2 \\
&= \frac{1}{|\mathcal{D}|} \sum_{\tau \in \mathcal{D}} \left( V_\theta(s_1) - \frac{r(\tau)}{\beta} + \sum_{h=1}^H \log \frac{\pi_\theta(a_h \mid s_h)}{\pi_{\mathsf{ref}}(a_h \mid s_h)} \right)^2 .
\end{aligned}
$$

A typical iterative variant of this approach can be formulated as:

$$\theta_{t+1} \leftarrow \underset{\theta}{\operatorname{argmin}} \, \frac{1}{|\mathcal{D}|} \sum_{\tau \in \mathcal{D}} \left( V_{\theta_t}(s_1) - \frac{r(\tau)}{\beta} + \sum_{h=1}^H \log \frac{\pi_\theta(a_h \mid s_h)}{\pi_{\mathsf{ref}}(a_h \mid s_h)} \right)^2 , \tag{14}$$

where $V_{\theta_t}$ is fixed from the previous iteration while optimizing for $\theta$.

To illustrate the difference between these approaches, we consider a simple 2-arm bandit problem where $r(a_1) = 1$ and $r(a_2) = 0$. We will show that, even at the population level, the iterative algorithm becomes trapped at a suboptimal solution, whereas TBRM converges to the globally optimal solution.

For this example, our Q-function class contains only two elements: $Q^\dagger = (0,0)$ and $Q^\star = (10,0)$, corresponding to the uniform policy and optimal policy, respectively.

Suppose at iteration $t$ we have $Q_t = Q^\dagger = (0,0)$, with temperature parameter $\beta = 0.1$, and data uniformly distributed over actions. The loss for the next iteration becomes:

$$\ell_t(Q) \coloneqq (1 - Q(a_1) + V_Q - V_t)^2 + (-Q(a_2) + V_Q - V_t)^2,$$

where $V_t = \beta \log \sum_a \exp(Q_t(a)/\beta)$ represents the value function from the current iteration.

In this setting, we can verify that $\ell_t(Q^\dagger) < \ell_t(Q^\star)$, meaning the iterative algorithm will select $Q_{t+1} = Q_t = Q^\dagger = (0,0)$ and remain stuck at this suboptimal solution. In contrast, $Q^\star$ is the global minimizer of the TBRM loss by definition, demonstrating the advantage of direct optimization over the iterative approach.

# G Implementation Details

## G.1 Training Details

To ensure rigorous and reproducible experimentation, we employ standardized and universally adopted hyperparameter settings, as detailed in Section 4.2. For baselines, we adhere closely to the recommended hyperparameter configurations as presented in VERL. Specifically, PPO training utilizes a learning rate of $1 \times 10^{-6}$ for the actor policy and $1 \times 10^{-5}$ for the critic policy. We incorporate a KL divergence coefficient and an entropy regularization coefficient of $0.001$ for PPO. The clip ratio for the actor loss function is set to $0.2$. For the GRPO baseline, we maintain the same KL divergence coefficient as PPO for the KL regularization term. To balance computational efficiency and performance, we utilize a prompt batch size of 128 and a maximum response length of 2048 tokens per training iteration. All generated responses are sampled using a temperature parameter of 1.0. For the TBRM method, the parameter $\beta$ is consistently set to $0.002$ across all experimental conditions. The learning rate for TBRM experiments is $2.5 \times 10^{-6}$, with the exception of the $\text{TBRM}_{n=1}$ with Qwen2.5-Math-7B model adopting a learning rate of $2 \times 10^{-6}$. All models are trained for a total of 760 steps.

All experiments are conducted on the same platform featuring 4x H100 80GB GPUs.

## G.2 Training Efficiency of TBRM

We compare the resource cost of TBRM, GRPO, and PPO in Section 4.3 by examining wall-clock time and maximal GPU memory usage. The labels for the time segments used in Figure 2(b) are detailed in Appendix G.2. To ensure a fair comparison, all experiments were conducted on the same platform featuring 4x H100 GPUs, and all configurations were standardized. Specifically, we employed vllm as the rollout backend and set `gpu_memory_utilization` to $0.4$. For policy updates, we set `micro_batch_size_per_gpu` to $1$, and for calculating log probabilities for both $\pi_{\theta_{\text{old}}}$ and $\pi_{\text{ref}}$, we also used a `micro_batch_size_per_gpu` of $1$. The value function model update for PPO also utilized a `micro_batch_size_per_gpu` of $1$. We use Qwen2.5-Math-7B as the base model, with a prompt batch size of 128.

| Name | Description | Involved Algorithms |
|---|---|---|
| ref_prob | Computing $\pi_{\text{ref}}(a_t \mid s_t)$ | TBRM, GRPO, PPO |
| old_prob | Computing $\pi_{\theta_{\text{old}}}(a_t \mid s_t)$ | GRPO, PPO |
| update_policy | Updating the policy parameter $\theta$ | TBRM, GRPO, PPO |
| ppo_update_value | Updating the value function model $V_\phi$ | PPO |
| ppo_value | Computing $V_\phi(s_t)$ | PPO |
| rollout | Sampling trajectories from the prompt set | TBRM, GRPO, PPO |
| other | Miscellaneous computations, e.g., rule-based reward $r(\tau)$, advantage (for GRPO and PPO), etc. | |

Table 3: Description of labels of time segments in Figure 2(b).

## G.3 Prompt Templates

**Qwen-Math Prompt Template.** We use the default prompt template of Qwen2.5-Math in the main experiments (Section 4.2).

```
<|im_start|>system
Please reason step by step, and put your final answer within \boxed{}.<|im_end|>
<|im_start|>user
question <|im_end|>
<|im_start|>assistant
```

**DeepSeek-R1 Prompt Template.** We use DeepSeek-R1 prompt template in the experiment discussed in Section 4.3.

```
A conversation between User and Assistant. The user asks a question, and the Assistant solves
it. The Assistant first thinks about the reasoning process in the mind and then provides the user
```

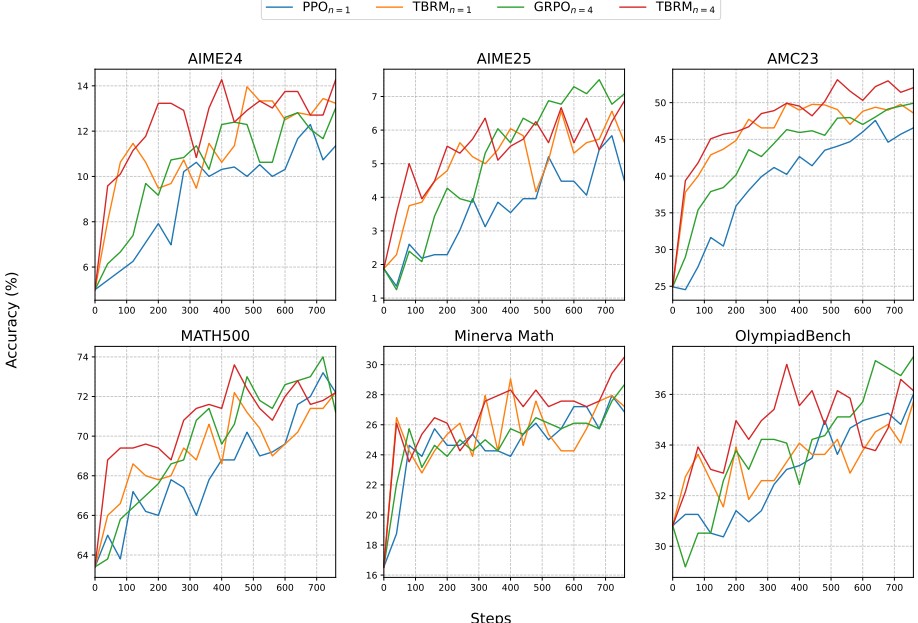

Figure 3: Learning accuracy across benchmarks with model Qwen2.5-Math-1.5B.

> with the answer. The reasoning process and answer are enclosed within `<think>` ... `</think>` and `<answer>` ... `</answer>` tags, respectively, i.e., `<think>` reasoning process here `</think>` `<answer>` answer here `</answer>`.
> User: **question**
> Assistant:

# H  Additional Experimental Results

## H.1  Training-Time Performance

Following the setup described in Section 4.1, we train and evaluate models of various sizes. Their performance on benchmarks during training is presented in Figure 3 and Figure 4.

## H.2  Qualitative Analysis

We delve into the responses of the TBRM model and observe intriguing reasoning behaviors. In this subsection, we highlight emergent reasoning patterns that the model develops during training, which contribute to its improved reasoning capabilities. Unless otherwise noted, all responses presented here are sampled from Qwen2.5-Math-7B-TBRM$_{n=4}$.

**Decomposition.**   The model demonstrates the ability to break down complex problems into smaller, manageable subproblems, solving each sequentially to construct the final solution. This hierarchical strategy reflects a structured and efficient approach to reasoning.

---

**Decomposition**

(MATH500) For what real values of $x$ is $-4 < x^4 + 4x^2 < 21$ satisfied? Express your answer in interval notation.

To solve the inequality $-4 < x^4 + 4x^2 < 21$, we need to break it into two parts and solve each part separately:
   1. Solve the inequality $x^4 + 4x^2 > -4$.
   2. Solve the inequality $x^4 + 4x^2 < 21$.
**Step 1: Solve** $x^4 + 4x^2 > -4$

---

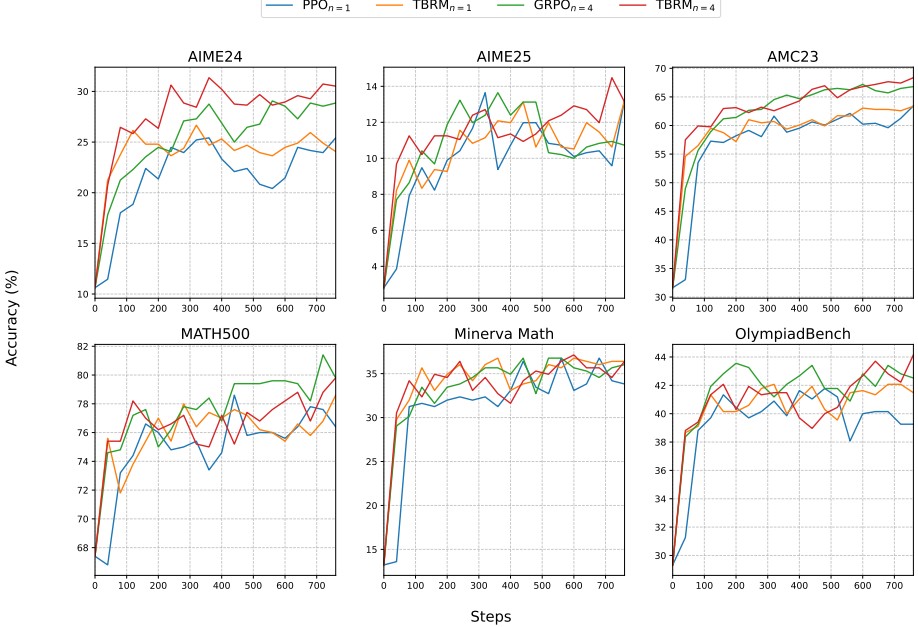

Figure 4: Learning accuracy across benchmarks with model Qwen2.5-Math-7B.

---

The expression $x^4 + 4x^2$ is always non-negative for all real numbers $x$ because both $x^4$ and $4x^2$ are non-negative. Therefore, $x^4 + 4x^2 \geq 0 > -4$ for all $x$. So, the inequality $x^4 + 4x^2 > -4$ is satisfied for all real numbers $x$.

**Step 2: Solve $x^4 + 4x^2 < 21$**

To solve $x^4 + 4x^2 < 21$, let's make a substitution. Let $y = x^2$. Then the inequality becomes $y^2 + 4y < 21$. Rearranging terms, we get $y^2 + 4y - 21 < 0$. We need to find the roots of the equation $y^2 + 4y - 21 = 0$ to determine the interval for $y$.

The quadratic equation $y^2 + 4y - 21 = 0$ can be factored as $(y + 7)(y - 3) = 0$. So, the roots are $y = -7$ and $y = 3$. Since $y = x^2$ and $x^2$ is always non-negative, we only consider the positive root $y = 3$. Thus, we need to solve $0 \leq y < 3$.

Substituting back $y = x^2$, we get $0 \leq x^2 < 3$. This means $-\sqrt{3} < x < \sqrt{3}$.

**Combining the Results**

Since the first inequality $x^4 + 4x^2 > -4$ is satisfied for all real numbers $x$, the solution to the original inequality $-4 < x^4 + 4x^2 < 21$ is determined by the second inequality. Therefore, the solution is $-\sqrt{3} < x < \sqrt{3}$.

The final answer in interval notation is:

$$\boxed{(-\sqrt{3}, \sqrt{3})}$$

---

**Enumeration.** The model adopts an exhaustive strategy by systematically iterating through possibilities or repeatedly applying a specific operation until the correct answer emerges. This approach is particularly useful in combinatorial or search-based problems.

---

**Enumeration**

(AIME24) Alice chooses a set $A$ of positive integers. Then Bob lists all finite nonempty sets $B$ of positive integers with the property that the maximum element of $B$ belongs to $A$. Bob's list has 2024 sets. Find the sum of the elements of A.

---

To solve this problem, we need to understand the relationship between the set $A$ and the number of nonempty subsets $B$ of positive integers where the maximum element of $B$ belongs to $A$.

If $A = \{a_1, a_2, \ldots, a_n\}$, then for each element $a_i \in A$, the number of subsets $B$ where the maximum element is $a_i$ is given by the number of nonempty subsets of $\{1, 2, \ldots, a_i - 1\}$, which is $2^{a_i - 1} - 1$. Summing over all elements in $A$, the total number of such subsets $B$ is:

$$(2^{a_1} - 1) + (2^{a_2} - 1) + \ldots + (2^{a_n} - 1) = 2^{a_1} + 2^{a_2} + \ldots + 2^{a_n} - n$$

Given that this total is 2024, we can write:

$$2^{a_1} + 2^{a_2} + \ldots + 2^{a_n} - n = 2024$$

or

$$2^{a_1} + 2^{a_2} + \ldots + 2^{a_n} = 2024 + n$$

To find the appropriate values for $a_1, a_2, \ldots, a_n$, we need to find a combination of powers of 2 that sum up to just over 2024. We can start by finding the largest power of 2 less than or equal to 2024. The largest power of 2 less than 2024 is $2^{10} = 1024$.
By subtracting 1024 from 2024, we get 1000. The next largest power of 2 less than or equal to 1000 is $2^9 = 512$. Subtracting 512 from 1000 gives 488. The next largest power of 2 less than or equal to 488 is $2^8 = 256$, leaving us with 232. The next largest power of 2 less than or equal to 232 is $2^7 = 128$, leaving us with 104. The next largest power of 2 less than or equal to 104 is $2^6 = 64$, leaving us with 40. The next largest power of 2 less than or equal to 40 is $2^5 = 32$, leaving us with 8. The next largest power of 2 less than or equal to 8 is $2^3 = 8$, leaving us with 0. So, the sum of the powers of 2 is:

$$1024 + 512 + 256 + 128 + 64 + 32 + 8 = 2024$$

This means that we have $n = 7$ elements in set $A$, and these elements are $\{11, 10, 9, 8, 7, 6, 4\}$ because the powers of 2 correspond to $2^{10}, 2^9, 2^8, 2^7, 2^6, 2^5, 2^3$.
So, the set $A = \{11, 10, 9, 8, 7, 6, 4\}$. The sum of the elements of $A$ is:

$$11 + 10 + 9 + 8 + 7 + 6 + 4 = 55$$

The final answer is:

$$\boxed{55}$$

**Verification.** The model exhibits self-monitoring behavior by verifying intermediate results during problem solving. This helps catch early mistakes and reinforces correct reasoning steps, enhancing both accuracy and reliability.

---

**Verification**

(MATH500) What is the length, in units, of the radius of a sphere whose volume and surface area, in cubic units and square units, respectively, are numerically equal?

To find the radius of a sphere whose volume and surface area are numerically equal, we need to set up the equations for the volume and surface area of a sphere and then solve for the radius $r$.
The volume $V$ of a sphere is given by: $V = \frac{4}{3}\pi r^3$
The surface area $A$ of a sphere is given by: $A = 4\pi r^2$
We are given that the volume and surface area are numerically equal, so we can set the two equations equal to each other: $\frac{4}{3}\pi r^3 = 4\pi r^2$
To solve for $r$, we can simplify the equation by dividing both sides by $4\pi r^2$ (assuming $r \neq 0$):
$\frac{r}{3} = 1$ $r = 3$
So, the radius of the sphere is $r = 3$. Let's verify this by substituting $r = 3$ back into the volume and surface area formulas to ensure they are indeed equal.
The volume when $r = 3$ is: $V = \frac{4}{3}\pi(3)^3 = \frac{4}{3}\pi \cdot 27 = 36\pi$
The surface area when $r = 3$ is: $A = 4\pi(3)^2 = 4\pi \cdot 9 = 36\pi$

Since the volume and surface area are both $36\pi$, our solution is correct. The length of the radius of the sphere is $\boxed{3}$.

**Backtracking.** Upon encountering contradictions or errors, the model is able to reassess its prior steps, revise incorrect assumptions, and pursue an alternative line of reasoning. This reflective capability mirrors human-like problem-solving resilience.

---

**Backtracking**

(MATH500) The asymptotes of a hyperbola are $y = 2x - 3$ and $y = 17 - 2x$. Also, the hyperbola passes through the point $(4, 7)$. Find the distance between the foci of the hyperbola.

To find the distance between the foci of the hyperbola, we need to follow these steps:

1. **Determine the center of the hyperbola:** The center of the hyperbola is the intersection point of the asymptotes. To find this point, we solve the system of equations given by the asymptotes:
$$y = 2x - 3$$
$$y = 17 - 2x$$
Setting the two equations equal to each other:
$$2x - 3 = 17 - 2x$$
Solving for $x$:
$$4x = 20 \implies x = 5$$
Substituting $x = 5$ back into one of the equations to find $y$:
$$y = 2(5) - 3 = 10 - 3 = 7$$
So, the center of the hyperbola is $(5, 7)$.

2. **Determine the slopes of the asymptotes:** The slopes of the asymptotes are $2$ and $-2$. This indicates that the hyperbola is oriented with its transverse axis vertical or horizontal. Since the slopes are $\pm 2$, the hyperbola is of the form:
$$\frac{(x-h)^2}{b^2} - \frac{(y-k)^2}{a^2} = -1 \quad \text{or} \quad \frac{(y-k)^2}{a^2} - \frac{(x-h)^2}{b^2} = 1$$
where the slopes of the asymptotes are $\pm \frac{a}{b}$ or $\pm \frac{b}{a}$. In this case, $\frac{a}{b} = 2$ or $\frac{b}{a} = 2$. Let's assume the hyperbola is of the form:
$$\frac{(y-7)^2}{a^2} - \frac{(x-5)^2}{b^2} = 1$$
with $\frac{a}{b} = 2$, so $a = 2b$.

3. **Use the point (4, 7) to find $a$ and $b$:** The hyperbola passes through the point $(4, 7)$. Substituting $x = 4$ and $y = 7$ into the equation of the hyperbola:
$$\frac{(7-7)^2}{a^2} - \frac{(4-5)^2}{b^2} = 1$$
Simplifying, we get:
$$0 - \frac{1}{b^2} = 1 \implies -\frac{1}{b^2} = 1 \implies b^2 = -1$$
This is incorrect, so let's assume the hyperbola is of the form:
$$\frac{(x-5)^2}{b^2} - \frac{(y-7)^2}{a^2} = -1$$
with $\frac{a}{b} = 2$, so $a = 2b$. Substituting $x = 4$ and $y = 7$:
$$\frac{(4-5)^2}{b^2} - \frac{(7-7)^2}{a^2} = -1$$
Simplifying, we get:
$$\frac{1}{b^2} - 0 = -1 \implies \frac{1}{b^2} = -1 \implies b^2 = 1$$
So, $b = 1$ and $a = 2b = 2$.

4. **Find the distance between the foci:** The distance between the foci of a hyperbola is $2c$, where $c = \sqrt{a^2 + b^2}$. In this case:

$$c = \sqrt{a^2 + b^2} = \sqrt{2^2 + 1^2} = \sqrt{4 + 1} = \sqrt{5}$$

So, the distance between the foci is:

$$2c = 2\sqrt{5}$$

The final answer is:

$$\boxed{2\sqrt{5}}$$

## H.3 Ablation Study: Classical BRM on LLMs

As discussed in Section 3, in the context of LLMs, directly minimizing BRM may not be possible because the token-level reward signal is either unavailable (e.g. if we assign the outcome reward to the final token) or very sparse. To highlight the advantages of the trajectory-level approach, we implemented classical BRM, whose loss function is defined in Eq. (3), using the same hyperparameters as in our main setup on Qwen2.5-Math-1.5B.

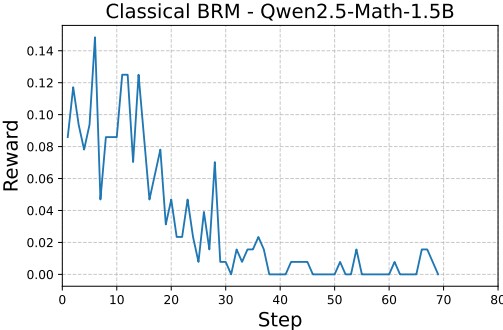

Figure 5: Training reward with classical BRM on model Qwen2.5-Math-1.5B.

Figure 5 shows that training reward quickly collapses, and we observe that the model outputs become random and meaningless. Intuitively, this degradation occurs because BRM has to propagate the sparse reward signal, which only receives at the final token, back through multiple token-wise regressions, whereas the TBRM provides a better implicit credit assignment through a single trajectory-level objective.

## H.4 Tasks Beyond Mathematical Problems

To demonstrate the generalizability of our method beyond mathematical tasks, we evaluate TBRM on five tasks from the reasoning-gym (Stojanovski et al., 2025) under the *graphs* category: course_schedule, family_relationships, largest_island, quantum_lock, and shortest_path. These tasks are naturally represented as graphs, consisting of nodes and edges, and typically require traversing connections to identify relationships, compute optimal paths, or determine reachable components. They involve reasoning patterns that differ significantly from those in mathematical tasks.

| Method | course_schedule | family_relationships | largest_island | quantum_lock | shortest_path | Average |
|---|---|---|---|---|---|---|
| **Qwen2.5-Math-1.5B** | 29.5 | 3.0 | 11.0 | 5.5 | 0.0 | 9.8 |
| GRPO $n = 4$ | 54.0 | **84.0** | 34.0 | **30.5** | 26.0 | 45.7 |
| TBRM $n = 4$ | **60.0** | 80.0 | **38.0** | 27.0 | **31.0** | **47.2** |

Table 4: Performance of GRPO and TBRM on various tasks from reasoning-gym, category *graphs*.

We construct a training set of 10,000 problems, with 2,000 questions per task, and a test set of 500 problems, comprising 100 questions from each task. For both training and evaluation, we use the official verifiers provided by reasoning-gym to compute rewards. Our experiments are conducted on

Qwen2.5-Math-1.5B using both TBRM and GRPO, with a prompt batch size of 1024 and 4 sampled responses per question ($n = 4$). Models are trained for 100 steps. All evaluations are conducted using greedy decoding. Results in Table 4 demonstrate that TBRM generalizes well to diverse reasoning tasks and performs on par with GRPO.

## H.5 TBRM with More Rollouts

To demonstrate that TBRM scales effectively with increasing number of sampled responses per prompt, we rerun GRPO and TBRM using most hyperparameters from DAPO (Yu et al., 2025). Specifically, we used a prompt batch size of 512 and generated $n = 16$ responses per prompt. For GRPO, we set the microbatch size to 512, resulting in 16 updates per training step. The experiments were conducted on the Qwen2.5-Math-7B model, following the same evaluation pipeline described in our paper. Both algorithms were trained for 100 steps. Table 5 shows that TBRM remains comparable to GRPO under these aligned settings.

| Method | AIME24 | AIME25 | AMC23 | MATH500 | Minerva Math | OlympiadBench |
| --- | --- | --- | --- | --- | --- | --- |
| | Avg@32 | Avg@32 | Avg@32 | Avg@1 | Avg@1 | Avg@1 |
| **Qwen2.5-Math-7B** | 10.6 | 2.8 | 31.6 | 67.4 | 13.2 | 29.3 |
| GRPO $n = 16$ | 26.6 | **11.0** | 61.8 | **77.8** | 32.7 | **40.4** |
| TBRM $n = 16$ | **27.9** | 10.9 | **62.8** | 76.4 | **33.5** | 39.9 |

Table 5: Performance of GRPO and TBRM with $n = 16$.

