# OpenReview forum: "Trajectory Bellman Residual Minimization: A Simple Value-Based Method for LLM Reasoning"
_NeurIPS.cc/2025/Conference — NeurIPS 2025 poster_

### Official Review · Reviewer_nMNz · 2025-06-27

**Clarity:** 3
**Significance:** 2
**Originality:** 3
**Rating:** 4
**Confidence:** 4

**Summary:**

The authors claim to provide a compelling alternative to policy gradient approaches that have dominated RL methods used in LLM applications. The approach presented is basically soft Q-learning lifted to the trajectory level by slightly modifying the classic Bellman residual minimization (BRM) objective. The authors justify the trajectory based variant of BRM (called TBRM) by proving that it optimizes the well-known KL-regularized RL reward objective. Finally experimental comparisons with PPO and GRPO are shown based on which the authors claim that TBRM consistently outperforms PPO and GRPO, indicating that value-based RL should be considered an alternative to policy gradient methods.

**Questions:**

Suggestions:
----

1) Address the theoretical equivalence with policy gradients. The discussion in lines 31-48 should be suitably updated.
2) Experiments:
-------
- If the claim is outperformance over policy gradient methods,
show results over more steps, multiple seeds to show variance, ablate against direct BRM optimization (standard soft-Q learning approach), compare with off-policy variants of soft-Q/PPO/GRPO, consider hyper-parameter sweep
- Alternatively, if only claiming comparable performance with methods like GRPO, highlight advantages of TBRM vs BRM with experiments.
- It would be good to see how the algorithm performance changes with more rollouts. The complexity vs performance tradeoff with GRPO would be nice to see.

**Ethical Concerns:**

["NO or VERY MINOR ethics concerns only"]

**Final Justification:**

It became clear during rebuttal that TRBM was comparable with GRPO, offering some ease of implementation and single-rollout use advantages. The theoretical guarantees and lifting the Bellman objective to trajectory level are novel and interesting directions. My borderline rating reflects the fact that in practice this method could be seen as an alternative to GRPO without providing a compelling enough reason or circumstance to switch.

**Limitations:**

No limitations are discussed. Some of the possible weaknesses discussed above could be worked in as limitations if not explicitly addressed.

**Quality:**

3

**Strengths And Weaknesses:**

Strengths:
-----------
1) The modification of the BRM to the TBRM objective and associated theoretical justification via convergence proof and relation to change of trajectory measure lemma is novel and interesting, but its significance is not evident (see discussion under weaknesses)
2) The single rollout performance of the algorithm seems promising, but low significance (see discussion under weaknesses)
3) The paper is well written, and technically sound.

Weaknesses:
-----------
1) The authors claim that the approach described is a compelling alternative that "consistently outperforms" policy gradient methods seems too strong for the following reasons:
- It is well known [Schulman et al., 2017a] that soft-Q learning with BRM objective is equivalent to a policy gradient algorithm (equivalence, modulo hyperparameter that weights critic term/policy term). Even TBRM backprops through the Q function into the policy (similar to DDPG as well). So the fact that it has good long horizon performance is not surprising. Could it be that TBRM performs well because it is in effect a policy gradient algorithm?
- The experimental results shown in Table 1 do not indicate clear outperformance over GRPO (n=4). Fig 1a shows GRPO quickly catching up, while Fig 1b. shows very close performance. This is consistent in Figure 2a and 2b. It would be good to see the results after several more steps. Also given the above point, the possibility that the perceived difference could be due to minor hyper-parameter settings cannot be ruled out.
- The fact that TBRM is off-policy is not a major point of differentiaton. Soft-Q based policy gradients can also easily expressed as off-policy algorithm (like DDPG). Off-policy versions of PPO[2, 3]  and theoretically grounded versions of GRPO exist as well [1] that show improved performance over on-policy versions. No comparisons with such methods are shown.
2) The main contribution seems to be the trajectory level loss formulation, yet no ablation against the classic BRM minimization objective is provided. This would help appreciate the advantage that TBRM brings ceteris paribus.
3) The significance of the single rollout (n=1) version seems limited. Memory usage is the same as for (n=4) but has lower compute time, one would probably prefer to spend the compute time (linear) for more rollouts to get the improved performance.

[1] Mroueh, Youssef, et al. "Revisiting Group Relative Policy Optimization: Insights into On-Policy and Off-Policy Training." arXiv preprint arXiv:2505.22257 (2025)

[2] Queeney, James, Yannis Paschalidis, and Christos G. Cassandras. "Generalized proximal policy optimization with sample reuse." Advances in Neural Information Processing Systems 34 (2021): 11909-11919.

[3] W. Meng, Q. Zheng, G. Pan, and Y. Yin. Off-policy proximal policy optimization. Proceedings of
the AAAI Conference on Artificial Intelligence, 37(8):9162–9170, 2023.

---

> ### Author Rebuttal · Authors · 2025-07-31
>
> Thank you for the review! Please see responses to individual questions below.
>
> **Q: Relation between TBRM and Policy Gradient**
>
> In comparison with policy gradient, the gradient of the TBRM loss can be decomposed as follows:
> $$
> \begin{align*}
>  & \nabla\_\theta \mathbb{E}\_{\tau \sim\mu}\left[\left(\mathsf{logit}\_\theta(s\_1,a\_1) - \sum\_{h=1}^{H}R\_\beta(s\_{h},a\_{h}) + \sum\_{h=2}^{H}\log\pi\_\theta(a\_{h} \mid s\_{h}) \right)^2 \right] \\\\
>  =& \nabla\_\theta \mathbb{E}\_{\tau\sim\mu}\left[\left(V\_\theta(s\_1) - \sum\_{h=1}^{H} R\_\beta(s\_{h},a\_{h}) + \sum\_{h=1}^{H} \log \pi\_\theta(a\_{h} \mid s\_{h}) \right)^2 \right] \\\\
> =& 2 \mathbb{E}\_{\tau \sim \mu} \left[ \left( V\_\theta(s\_1) - \sum\_{h=1}^{H} R\_\beta(s\_{h},a\_{h}) + \sum\_{h=1}^{H} \log \pi\_\theta(a\_{h} \mid s\_{h}) \right) \cdot \left(\sum\_{h=1}^{H} \nabla\_\theta \log \pi\_\theta(a\_{h} \mid s\_{h}) + \nabla\_\theta V\_\theta (s\_1) \right)
> \right] \\\\
> =& \mathbb{E}\_{\tau \sim \mu} \Bigg[ - \left( \sum\_{h=1}^{H} R\_\beta(s\_{h},a\_{h}) -  \sum\_{h=1}^{H} \log \pi\_\theta(a\_{h} \mid s\_{h}) - V\_{\theta}(s\_1)  \right) \sum\_{h=1}^{H} \nabla\_\theta \log \pi\_\theta(a\_{h} \mid s\_{h}) \\\\
> &+\nabla\_\theta \left( V\_\theta(s\_1) - \sum\_{h=1}^{H} R\_\beta(s\_{h},a\_{h}) + \sum\_{h=1}^{H} \log \pi\_{\overline{\theta}}(a\_{h} \mid s\_{h}) \right)^2 \Bigg] \\\\
> \end{align*}
> $$
> Where $\mu$ is the policy used to generate the trajectories, and $\overline{\theta}$ denotes a copy of $\theta$ whose parameters are held fixed (i.e., not updated). In the on-policy case, i.e., $\mu = \pi_\theta$, the first term in TBRM resembles the REINFORCE gradient but with a baseline $V_\theta(s_1)$ for variance reduction, and the second term corresponds to the gradient of the value function.
>
> **Q: Comparison between the Performance of TBRM and GRPO**
>
> We acknowledge that the performance of TBRM is comparable to GRPO. The main advantage of TBRM is that: it is simpler to implement, requiring neither importance-sampling ratios nor clipping; it is more flexible, as it naturally supports off-policy training and works effectively with a single rollout; and it provides theoretical guarantees, including finite-sample bounds. These strengths make TBRM a practical and principled alternative across a range of scenarios, maybe particularly in tasks involving multi-turn conversations and agentic workflows, where trajectory collection is both costly and challenging. We view adapting TBRM to such settings as a promising direction for future work.
>
> In the next revision of our paper, we will revise the wording that suggests TBRM consistently outperforms GRPO and instead emphasize that the two methods achieve comparable performance. This clarification is intended to prevent potential confusion for readers.
>
> **Q: Ablation Study of State-Action-Level BRM**
>
> We agree with the reviewer on the importance of including an ablation study compared against the original BRM. To highlight the advantages of the trajectory-level approach, we implemented a state-action-level variant of BRM and ran experiments using the same hyperparameters as in our main setup on Qwen2.5-Math-1.5B. The results show that training reward quickly collapses, and the model outputs become random and meaningless. Intuitively, this degradation occurs because BRM has to propagate the sparse reward signal, which only receives at the final token, back through multiple token-wise regressions, whereas the TBRM provides a better implicit credit assignment through a single trajectory-level objective.
>
> | Step |0|5|10|15|20|25|30|35|40|
> |-|-|-|-|-|-|-|-|-|-|
> | Reward |0.0859|0.1484|0.1250|0.0469|0.0234|0.0391|0.0000|0.0234|0.0000|
>
> **Q: Experiment Results with More Rollouts**
>
> We rerun GRPO and TBRM using most hyperparameters from DAPO [1] to ensure a fair comparison. Specifically, we used a prompt batch size of 512 and generated n=16 responses per prompt. For GRPO, we set the microbatch size to 512, resulting in 16 updates per training step. The experiments were conducted on the Qwen2.5-Math-7B model, following the same evaluation pipeline described in our paper. Both algorithms were trained for 100 steps. Each run takes approximately 40 hours on 4x H100 GPUs. Our results show that TBRM remains comparable to GRPO under these aligned settings.
>
> | | AIME24 | AIME25 | AMC23 | MATH500 | Minerva Math | OlympiadBench |
> |-|-|-|-|-|-|-|
> | Qwen2.5-Math-7B | 10.6%| 2.8% | 31.6% | 67.4% | 13.2% | 29.3%|
> | GRPO (n=16)| 26.6%| 11.0%| 61.8% | 77.8% | 32.7%| 40.4%|
> | TBRM (n=16)| 27.9%| 10.9%| 62.8% | 76.4% | 33.5% | 39.9%|
>
> ---
> [1] DAPO: An Open-Source LLM Reinforcement Learning System at Scale. Yu et al. 2025

---

> > ### Comment · Reviewer_nMNz · 2025-08-05
> >
> > I thank the authors for the rebuttal and providing additional experiments.  The experiments show, and the authors acknowledge, that the performance of TBRM was comparable to GRPO.
> >
> > The authors call out the key advantages of TBRM: "is simpler to implement, requiring neither importance-sampling ratios nor clipping; it is more flexible, as it naturally supports off-policy training and works effectively with a single rollout; and it provides theoretical guarantees".
> >
> > 1) Simpler to implement: no importance ratios/clipping is a minor point
> > 2) Supporting off-policy learning: There are off-policy alternatives of PPO and GRPO (references cited in original review) that were not compared against, so it is hard to say how significant this aspect of TRBM is.
> > 3) Works with single rollout: I consider this a minor benefit (please see original review for justification)
> > 4)  Theoretical guarantees: Based on the current paper, I would regard this as the most significant contribution of TBRM. I appreciate the novel theoretical insight and direction.
> >
> > On balance, however, I must keep my borderline assessment, since the theoretical analysis did not translate to clear experimental impact.

---

### Official Review · Reviewer_3Y8d · 2025-07-05

**Clarity:** 4
**Significance:** 3
**Originality:** 3
**Rating:** 5
**Confidence:** 3

**Summary:**

This work proposes a new approach to RL finetuning of LLMs. Whereas previous approaches (PPO, GRPO) are policy gradient methods, the authors propose TBRM, a a q-learning method with similarities to soft q-learning. The models output logits are treated as q-values. The authors leverage a soft approximation to the maximum value function (softmax value) to create a bellman equation. This is then turned into a loss that optimizes the trajectory instead of the step-wise as usual. Under certain assumptions, they show that this algorithm likely converges close to an optimal policy within a finite number of samples. They then train Qwen2.5-Math base models (1.5B and 7B) on DAPO's math problem dataset and show improvements over GRPO on a variety of standard benchmarks (AIME, Minerva, MATH500, Olympiad).

**Questions:**

**major questions**

What does the loss for your method look like in practice in code? E.g. is `logit(s1,a1)` simply the logit corresponding to `a1` in the transformer after the prompt? This is what algorithm 1 should be, as it is currently just explaining what on-policy RL is without really any specifics to your method.

How does your loss compare to soft q learning and regular policy gradient methods. Especially given Schulman et al's equivalence between Soft Q-learning and Policy gradient can you show how the gradient of your method differs from a regular REINFORCE policy gradient? Ideally, also compare to [CoPG](https://aclanthology.org/2024.emnlp-main.1190.pdf) (Flet-Berliac, Grinsztajn et al, 2024) as it also claims to be off-policy, doesn't use importance sampling, and is based on mirror-descent which has connections to your work.

**better benchmarking**

Can you compare using DAPO's hyperparameters setting for their dataset i.e. K=16 responses per prompt, batch size 512, etc. You don't need to do 16 minibatches per training step with your 8 GPU setup, 1-2 will likely be sufficient. They used Qwen2.5 32B but keeping Qwen2.5 7B is sufficient, just using their hyperparams for GRPO / DAPO is a good baseline and tests your claims of off-policyness.

**clarification**

I assume by `logit(s,a)` you implement this as inputting the state (previous tokens) and take the unnormalized logit corresponding to token `a`. Is this correct?

please define your q-value $Q(s,a)$ as used in the equations after equation 1 on page 4. It would help for readability if it were included here

**nitpicks**

In modern deep learning papers, generally `softmax` refers to the soft argmax. You're using it in the other sense and for clarify maybe `soft approximation to max` or `logsumexp` may be clearer. At least a footnote with clarification is helpful.

**Ethical Concerns:**

["NO or VERY MINOR ethics concerns only"]

**Final Justification:**

I think this work demonstates that q-learning methods can indeed work at the highest level for RL for LLMs and finds elegant theory that supports their claim. Though they don't outperform GRPO, I wouldn't necessarily expect a first attempt at q-learning to beat a refined method like GRPO, especially when the base model was likely trained to be receptive to that specific RL finetuning. I think this is a fine paper and would make a good impact in the community.

**Limitations:**

yes

**Quality:**

2

**Strengths And Weaknesses:**

**strengths**

The idea of applying q-learning to RL for LLMs is both timely and interesting. The approach of using the pure logits as q-values is quite bold (previous works like SimPO have been quite unstable) and interesting. The theoretical guarantees have quite some caveats but a finite sample guarantee is interesting.

Overall the paper is quite well written and mostly clear with the math. The theory generally tracks with soft q-learning and  The empirical setup is good and demonstrates reasonable improvement over GRPO. The attention to memory and compute efficiency is great. The idea of q-learning approaches that can be off-policy is quite useful especially in current async RLHF setups.


**weaknesses**

The **major weakness** of the paper is the lack of comparison to related losses in previous literature

First and foremost, the authors do not compare their loss or the gradient of their loss to previous methods like soft q-learning and to standard policy gradient + KL losses. This is especially important as the loss in equation 3 can be rearranged to become $(V(s_1) - r(\tau) / \beta + KL(\pi | \pi_{ref}))^2$ which has a lot of similarities to both soft q learning and policy gradient with KL. Given that Schulman et al (2018) showed an equivalence between these two, it is important to demonstrate how this new q-learning loss relates to previous losses and what it does differently.


There are **minor weaknesses** related to the relevance of the theory and overselling

It seems like turning a step into trajectory bellman equation is trivial for LLMs, please correct me if I'm wrong. In current LLMs, the transition function $s,a \to s'$ is just adding the current token $a_{t+1}$ to the previous tokens $s = a_1 \ldots a_t$. Therefore, you can simply treat RL for LLMs not as an MDP but a contextual bandit and a single step action.

The theory that TBRM requires only 1 sample per prompt is clearly not going to work in practice on sufficiently difficult datasets. The likelihood of sampling a correct answer on a difficult prompt is very low and many datasets will have pass@1 of 0. Modern methods sample upwards of 64 responses per prompt to get any signal at all. Experiments here start with pass@1 accuracy of ~25% which is not necessarily realistic.


**overall**

This work presents an interesting q-learning approach to RL finetuning and results are promising. I think it is currently in a state that is just over the bar of acceptance but not very strong. If (1) the practical implementation of the loss is clarified and its calculation is explicitly stated and (2) it is explained how the loss compares to existing REINFORCE + KL methods (and ideally CoPG), I will happily increase the score to accept and argue for it. If the authors also do experiments replicating DAPO's hyperparamters in a 7B setup and find good results, I would increase my score to a strong accept.

---

> ### Author Rebuttal · Authors · 2025-07-31
>
> Thank you for the review! Please see responses to individual questions below.
>
> **Q: Comparison with Other Related Algorithms**
>
> For a detailed comparison between TBRM and soft Q-learning, we refer the reviewer to the updated Appendix E of the supplementary material (please see Appendix E in the ZIP file of the supplementary material, which has been updated from the version in the main paper). The major differences include (1) TBRM employs direct optimization rather than an iterative approach, and (2) TBRM’s loss function operates on complete trajectories rather than summing losses over individual timesteps within trajectories, hence eliminating the need for per-step reward.
>
> In comparison with policy gradient, the gradient of the TBRM loss can be decomposed as follows:
> $$
> \begin{align*}
>  & \nabla\_\theta \mathbb{E}\_{\tau \sim\mu}\left[\left(\mathsf{logit}\_\theta(s\_1,a\_1) - \sum\_{h=1}^{H}R\_\beta(s\_{h},a\_{h}) + \sum\_{h=2}^{H}\log\pi\_\theta(a\_{h} \mid s\_{h}) \right)^2 \right] \\\\
>  =& \nabla\_\theta \mathbb{E}\_{\tau\sim\mu}\left[\left(V\_\theta(s\_1) - \sum\_{h=1}^{H} R\_\beta(s\_{h},a\_{h}) + \sum\_{h=1}^{H} \log \pi\_\theta(a\_{h} \mid s\_{h}) \right)^2 \right] \\\\
> =& 2 \mathbb{E}\_{\tau \sim \mu} \left[ \left( V\_\theta(s\_1) - \sum\_{h=1}^{H} R\_\beta(s\_{h},a\_{h}) + \sum\_{h=1}^{H} \log \pi\_\theta(a\_{h} \mid s\_{h}) \right) \cdot \left(\sum\_{h=1}^{H} \nabla\_\theta \log \pi\_\theta(a\_{h} \mid s\_{h}) + \nabla\_\theta V\_\theta (s\_1) \right)
> \right] \\\\
> =& \mathbb{E}\_{\tau \sim \mu} \Bigg[ - \left( \sum\_{h=1}^{H} R\_\beta(s\_{h},a\_{h}) -  \sum\_{h=1}^{H} \log \pi\_\theta(a\_{h} \mid s\_{h}) - V\_{\theta}(s\_1)  \right) \sum\_{h=1}^{H} \nabla\_\theta \log \pi\_\theta(a\_{h} \mid s\_{h}) \\\\
> &+\nabla\_\theta \left( V\_\theta(s\_1) - \sum\_{h=1}^{H} R\_\beta(s\_{h},a\_{h}) + \sum\_{h=1}^{H} \log \pi\_{\overline{\theta}}(a\_{h} \mid s\_{h}) \right)^2 \Bigg] \\\\
> \end{align*}
> $$
> Where $\mu$ is the policy used to generate the trajectories, and $\overline{\theta}$ denotes a copy of $\theta$ whose parameters are held fixed (i.e., not updated). In the on-policy case, i.e., $\mu = \pi_\theta$, the first term in TBRM resembles the REINFORCE gradient but with a baseline $V_\theta(s_1)$ for variance reduction, and the second term corresponds to the gradient of the value function.
>
> Finally, CoPG is basically optimizing the difference between the Bellman residuals of two trajectories. It falls under the category of “DAA-pair” algorithms discussed in Appendix E (please see Appendix E in the ZIP file of the supplementary material, which has been updated from the version in the main paper).
>
> **Q: Modeling LLM Rollout as MDP or Contextual Bandit**
>
> Our setting applies to more general deterministic MDPs than the token-level MDP, even when the transition dynamics are unknown a priori. This formulation naturally captures a wide range of scenarios beyond standard single-turn question answering. Notable examples include multi-turn conversations and agentic workflows [1], which cannot be adequately modeled as contextual bandits. For instance, outputs from external tools are part of the deterministic state transitions, so the transition is no longer merely appending new actions to the state. Our algorithm, with MDP setting, correctly avoids taking gradients on the output from external tools, whereas a naive contextual bandit approach would incorrectly take gradients through them. This limitation of contextual bandits has been empirically demonstrated in [2].
>
> Additionally, while token-level MDPs are statistically equivalent to contextual bandits in some cases, contextual bandit algorithms may be computationally inefficient when applied to token-level settings, for example, when they require explicit optimization over the action space, which corresponds to all possible token sequences.
>
> **Q: Limitations of Single-Sample TBRM on Challenging Datasets**
>
> We agree that on sufficiently challenging datasets, TBRM may require multiple samples to achieve strong performance. However, our claim that "TBRM only requires one sample per prompt" refers to the flexibility of its objective formulation, rather than its statistical efficiency. We will clarify this distinction in the next revision to avoid confusion.
>
> **Q: Better Benchmarking with Hyperparameters from DAPO**
>
> We rerun GRPO and TBRM using most hyperparameters from DAPO [3] to ensure a fair comparison. Specifically, we used a prompt batch size of 512 and generated n=16 responses per prompt. For GRPO, we set the microbatch size to 512, resulting in 16 updates per training step. The experiments were conducted on the Qwen2.5-Math-7B model, following the same evaluation pipeline described in our paper. Both algorithms were trained for 100 steps. Each run takes approximately 40 hours on 4x H100 GPUs. Our results show that TBRM remains comparable to GRPO under these aligned settings.
>
> | | AIME24 | AIME25 | AMC23 | MATH500 | Minerva Math | OlympiadBench |
> |-|-|-|-|-|-|-|
> | Qwen2.5-Math-7B | 10.6%| 2.8% | 31.6% | 67.4% | 13.2% | 29.3%|
> | GRPO (n=16)| 26.6%| 11.0%| 61.8% | 77.8% | 32.7%| 40.4%|
> | TBRM (n=16)| 27.9%| 10.9%| 62.8% | 76.4% | 33.5% | 39.9%|
>
> **Q: Clarification on logits used in the algorithm**
>
> $\mathsf{logit}_\theta (s_1,a_1)$ is the raw logit (before softmax) of the first output token with input being a prompt from the dataset.
>
> We agree with the reviewer that providing a more precise description of the algorithm and a clearer definition of the Q-function would improve readability. We will revise the wording accordingly in our next revision.
>
> ---
> [1] Search-r1: Training llms to reason and leverage search engines with reinforcement learning. Jin et al. 2025
>
> [2] Building math agents with multi-turn iterative preference learning. Wei et al. 2024
>
> [3] DAPO: An Open-Source LLM Reinforcement Learning System at Scale. Yu et al. 2025

---

> > ### Author Response · Authors · 2025-08-08
> >
> > Dear Reviewer 3Y8d,
> >
> > As the author–reviewer discussion window is nearing its close, we would be very grateful to hear whether our updates have addressed your concerns.
> >
> > Summary of updates in our rebuttal:
> > - **Loss comparison**: Compared our algorithm’s loss with that of related methods.
> > - **MDP vs. contextual bandits**: Explained why MDP modeling offers advantages over contextual bandits.
> > - **Reproduction with DAPO settings**: Reran both the baseline and our method using settings closely matching DAPO’s original configuration.
> > - **Notation and one-rollout claim**: Clarified our notation and the one-rollout claim.
> >
> > We truly appreciate the time you have spent reviewing our work, and we are happy to provide any further clarification or details if you have remaining questions.
> >
> > Best,
> > Authors

---

> > > ### Comment · Reviewer_3Y8d · 2025-08-08
> > >
> > > Thank you for the clear and useful rebuttal. I believe this paper should be accepted to the conference and will update my review to reflect this.
> > >
> > > **loss comparison** this is thorough and very useful! I like Appendix E and think it is a great addition to the paper
> > >
> > > **mdp vs contextual bandit** the clarification about multi-turn and tool-calling scenarios makes sense, thank you
> > >
> > > **DAPO reproduction** is great and very useful, thank you. I didn't expect improvements over GRPO but matching it is notable and demonstrates TBRM as a strong alternative to GRPO and policy gradient in general.

---

### Official Review · Reviewer_ufKS · 2025-07-06

**Clarity:** 3
**Significance:** 3
**Originality:** 3
**Rating:** 4
**Confidence:** 4

**Summary:**

This paper adapts the paradigm of Bellman Residual Minimization over to the LLM reasoning setting. They use the LLM’s logits as Q-values, and operate with a single rollout per prompt. They adapt the Bellman Residual Formulation to require only terminal rewards (no intermediate value estimates)--leveraging the deterministic nature of the LLM reasoning chain MDP. Experiments on mathematical-reasoning benchmarks show that performance can exceed PPO and GRPO.

**Questions:**

I tried to investigate the contents of the training set from DAPO, published in 2025, but was unable to figure out whether there is train-test overlap with the various math benchmarks in Table 1 (some of which are several years old). I have some concerns about train-test overlap in the tasks tested—but would appreciate any clarification the authors can offer on this.

Would also appreciate more clarification on the advantage of TBRM with n=4 as compared to GRPO with n=4—and I think it would probably be good to signpost in the introduction that best TBRM performance is obtained with n>1 since the paper claims multiple rollouts are not necessary (strictly true, but not optimal performance)

**Ethical Concerns:**

["NO or VERY MINOR ethics concerns only"]

**Final Justification:**

I appreciate the novelty of TBRM and the mathematical foundations as well as the new experiments across a second type of dataset--but we seem unclear on the relative performance of TBRM and GRPO as model size scales, and I'm definitely concerned about the provenance of the DAPO dataset. Overall, I think the mathematical foundations, simplicity, and experiments across two settings justify acceptance.

**Limitations:**

Yes

**Quality:**

3

**Strengths And Weaknesses:**

**Strengths**

The mathematical foundations of the algorithm are principled, and the idea of using the LLM’s logits as Q-values are novel and interesting.

I particularly appreciated the simplicity and effectiveness of the formulation.
By removing significant algorithmic complexity vs PPO/GRPO (clipping, multiple rollouts, etc.), the algorithm has the potential to be more scalable than competing approaches.

Despite only needing a single rollout per prompt, TBRM also shows benefits from having multiple rollouts per prompt—similar to GRPO.

**Weaknesses**

For the larger LLM tested—Qwen 2.5 7B, performance of GRPO and TBRM seems comparable at training time in the n=4 setting (and TBRM with n=1 is worse). This suggests that the two approaches may be comparable for larger models (and both have equal complexity?) Memory usage of both approaches in Figure 2B also look comparable. More clarity on this front would be appreciated :)

I would have appreciated seeing experiments on a second domain beyond math--and I am struggling to figure out the exact contents of the train set from DAPO. In the current writing the reader doesn't have the information necessary to understand how the test tasks relate to the contents of the training set.

---

> ### Author Rebuttal · Authors · 2025-07-31
>
> Thank you for the review! Please see responses to individual questions below.
>
> **Q: Comparison between TBRM and GRPO with $n=4$**
>
> We acknowledge that TBRM and GRPO achieve comparable performance on Qwen2.5-Math-7B when using 4 rollouts. However, TBRM offers several notable advantages: it is simpler to implement, avoiding the need for importance sampling ratios or clipping; it is more flexible, naturally accommodating off-policy training and performing effectively even with a single rollout; and it is theoretically grounded, providing finite-sample guarantees. These properties make TBRM a practical and principled choice across a wide range of settings.
>
> To prevent potential confusion, we will clarify in the introduction that TBRM achieves its best performance when using more than one rollout ($n > 1$).
>
> **Q: Extension to Non-Mathematical Tasks**
>
> To demonstrate the generalizability of our method beyond mathematical tasks, we evaluate it on five tasks from the reasoning-gym [1] under the "graphs" category: course_schedule, family_relationships, largest_island, quantum_lock, and shortest_path. These tasks are naturally represented as graphs, consisting of nodes and edges, and typically require traversing connections to identify relationships, compute optimal paths, or determine reachable components. They involve reasoning patterns that differ significantly from those in mathematical tasks.
>
> We construct a training set of 10,000 problems, with 2,000 questions per task, and a test set of 500 problems, comprising 100 questions from each task. For both training and evaluation, we use the official verifiers provided by reasoning-gym to compute rewards. Our experiments are conducted on Qwen2.5-Math-1.5B using both TBRM and GRPO, with a prompt batch size of 1024 and 4 sampled responses per question (n = 4). Models are trained for 100 steps. All evaluations are conducted using greedy decoding. Results demonstrate that TBRM generalizes well to diverse reasoning tasks and performs on par with GRPO.
>
> | | course_schedule | family_relationships | largest_island | quantum_lock | shortest_path | Average |
> |-|-|-|-|-|-|-|
> | Qwen2.5-Math-1.5B | 29.5% | 3.0%| 11.0%| 5.5% | 0.0% | 9.8%|
> | GRPO (n=4) | 54.0% | 84.0% | 34.0%| 30.5%| 26.0%| 45.7% |
> | TBRM (n=4) | 60.0% | 80.0% | 38.0%| 27.0%| 31.0%| 47.2% |
>
> **Q: Clarification on Training Set DAPO**
>
> We thank the reviewer for raising this important point. Our training data is an open-source dataset (DAPO [2]), which is widely used in the community. We did not curate or modify this training set, and our work does not aim to contribute to dataset construction. Instead, our focus is on the methodological advancement evaluated on standard benchmarks.
>
> To mitigate concerns about potential train-test leakage, we selected well-established benchmarks as our test sets, which are publicly available and commonly used for evaluation. However, since we rely on existing open-source data for training, we acknowledge that we cannot guarantee the absence of any incidental overlap (e.g., paraphrased or similar examples) without access to the full provenance of the dataset.
>
> ---
> [1] REASONING GYM: Reasoning Environments for Reinforcement Learning with Verifiable Rewards. Stojanovski et al. 2025
>
> [2] DAPO: An Open-Source LLM Reinforcement Learning System at Scale. Yu et al. 2025

---

> > ### Comment · Reviewer_ufKS · 2025-08-05
> >
> > Thank you for your rebuttal.
> >
> > I appreciate the comments on the relative practicality of TBRM--as it is certainly a reasonable consideration for practitioners, though it would definitely be preferable to offer improved performance as well to motivate adoption in existing settings where GRPO is used.
> >
> > Thank you for the additional experiment on reasoning-gym--the additional non-mathematical benchmark is quite valuable to show the generalization of the algorithm to a new setting.
> >
> > I will note that the DAPO dataset was released in March 2025, meaning it was no more than 2 months old at the time of this submission--so I think it is a bit alarming that we are accepting the dataset is "widely used" and "commonly used for evaluation" without knowledge of the provenance of the dataset. To me, it's not the same as demonstrating experimental results ex. on ImageNet for a computer vision paper from a few years ago. Even setting aside concerns about when RL is "sharpening" model outputs vs introducing new capabilities, it seems particularly important to know at least whether or not there is overlap between the RL training and test sets.
> >
> > I appreciate the novelty of TBRM and the mathematical foundations as well as the new experiments across a second type of dataset--but we seem unclear on the relative performance of TBRM and GRPO as model size scales, and I'm definitely concerned about the provenance of the DAPO dataset. On balance, I will maintain my score.

---

### Official Review · Reviewer_nij4 · 2025-07-09

**Clarity:** 3
**Significance:** 3
**Originality:** 3
**Rating:** 4
**Confidence:** 3

**Summary:**

This paper introduces Trajectory Bellman Residual Minimization (TBRM), a novel value-based reinforcement learning method for large language models. Departing from previous policy-based approaches like PPO and GRPO, TBRM minimizes a trajectory-level Bellman residual by leveraging the model’s logits as Q-values, thus requiring only one rollout per prompt. Key contributions include: (1) a simple, off-policy algorithm eliminating critics, importance sampling, and clipping mechanisms; (2) theoretical proof of convergence to the near-optimal KL-regularized policy via an improved change-of-measure analysis; (3) empirical validation across six mathematical reasoning benchmarks (e.g., AIME24, MATH500), where TBRM outperforms baselines by a large margin.

**Questions:**

- Equation 3 is confusing becase of interleaving use of $\texttt{logit}$ and $\pi_\theta$. After simplification, Equation 3 looks like the $L_2$ loss version of AWR [1]. Can the authors clarify on this?
- Will you add error bars or confidence intervals to Table 1/Figure 1 to strengthen reproducibility?
- (Minor) Could TBRM extend to non-mathematical tasks (e.g., ALFWorld) where reasoning patterns differ? Preliminary results on one agentic benchmark would highlight versatility.



[1] Peng, Xue Bin, et al. "Advantage-weighted regression: Simple and scalable off-policy reinforcement learning." arXiv preprint arXiv:1910.00177 (2019).

**Ethical Concerns:**

["NO or VERY MINOR ethics concerns only"]

**Final Justification:**

Most of my concerns are resolved during the rebuttal period. However, I still have concerns about the links between the theoretical insights and strong experimental results. Therefore, I recommend weak acceptance.

**Limitations:**

yes

**Quality:**

3

**Strengths And Weaknesses:**

**Strengths**
- Clarity. The paper is clearly writen, with structured preliminaries and intuitive algorithm derivation. Notation is consistent, and figures/tables effectively support claims.
- Significance. TBRM addresses critical limitations of policy-based methods and one-rollout efficiency offer practical value for scaling RL in LLMs.
- Theoretical soundness. The convergence proof under off-policy data and deterministic MDPs strengthens validity of the proposed method.

**Weaknesses**
- The proof is done with deterministic MDPs. This makes the theoretical result less significant and making the algorithm less practical in environments with randomness (e.g., Embodied Agents, GUI Agents, etc.)
- Results lack error bars/confidence intervals (e.g., Figure 1 and Table 1), limiting assessment of performance variance across runs.

---

> ### Author Rebuttal · Authors · 2025-07-31
>
> Thank you for the review! Please see responses to individual questions below.
>
> **Q: Limitations of Deterministic MDPs in Algorithmic Proofs**
>
> Our algorithm is designed for discrete, tokenized settings relevant to language models. While the guarantee for Algorithm 1 (Theorem 2) only works for deterministic MDPs, the change-of-trajectory-measure lemma (Lemma 1) holds for stochastic MDPs as well. And this lemma has broader use beyond the analysis of the algorithm in this paper. Extending our algorithm and its analysis to the general MDP setting is beyond the scope of this work and is left for future research.
>
> **Q: Missing Error Bars/Confidence Interval**
>
> We rerun the evaluation five times with different random seeds and compute confidence intervals following the methodology used in arena-hard-auto [1]. We report the 90% confidence interval, using the 0.05 and 0.95 quantiles as the lower and upper bounds, respectively.
>
> |**Model**| **AIME24** | **AIME24_ci** | **AIME24_ci** | **AIME25** | **AIME25_ci** | **AIME25_ci** | **AMC23** | **AMC23_ci** | **AMC23_ci** | **Math500** | **Math500_ci** | **Math500_ci** | **Minerva Math** | **Minerva Math_ci** | **Minerva Math_ci** | **OlympiadBench** | **OlympiadBench_ci** | **OlympiadBench_ci** |
> |-|-|-|-|-|-|-|-|-|-|-|-|-|-|-|-|-|-|-|
> || Avg| 05| 95| Avg| 05| 95| Avg | 05 | 95 | Avg | 05 | 95 | Avg| 05| 95| Avg | 05 | 95 |
> | **Qwen2.5-Math-1.5B** | 0.051| 0.046 | 0.056 | 0.014| 0.011 | 0.017 | 0.250 | 0.242| 0.258| 0.635 | 0.620| 0.647| 0.165| 0.149 | 0.182 | 0.304 | 0.291| 0.318|
> | **PPO-1.5B-n=1**| 0.113| 0.105 | 0.120 | 0.050| 0.045 | 0.056 | 0.457 | 0.446| 0.467| 0.708 | 0.695| 0.722| 0.268| 0.251 | 0.288 | 0.361 | 0.345| 0.374|
> | **TBRM-1.5B-n=1** | 0.128| 0.120 | 0.137 | 0.064| 0.057 | 0.069 | 0.500 | 0.489| 0.511| 0.724 | 0.709| 0.736| 0.271| 0.251 | 0.288 | 0.341 | 0.327| 0.354|
> | **GRPO-1.5B-n=4** | 0.129| 0.120 | 0.136 | 0.078| 0.070 | 0.084 | 0.498 | 0.489| 0.508| 0.704 | 0.689| 0.719| 0.287| 0.268 | 0.307 | 0.369 | 0.356| 0.381|
> | **TBRM-1.5B-n=4** | 0.139| 0.132 | 0.146 | 0.065| 0.059 | 0.070 | 0.534 | 0.524| 0.545| 0.714 | 0.701| 0.726| 0.302| 0.284 | 0.327 | 0.371 | 0.355| 0.387|
> | **Qwen2.5-Math-7B** | 0.100| 0.093 | 0.109 | 0.034| 0.030 | 0.038 | 0.308 | 0.298| 0.317| 0.675 | 0.661| 0.690| 0.131| 0.115 | 0.144 | 0.293 | 0.280| 0.304|
> | **PPO-7B-n=1**| 0.240| 0.233 | 0.249 | 0.122| 0.113 | 0.127 | 0.628 | 0.617| 0.638| 0.764 | 0.749| 0.776| 0.338| 0.321 | 0.360 | 0.394 | 0.382| 0.407|
> | **TBRM-7B-n=1** | 0.250| 0.240 | 0.258 | 0.123| 0.115 | 0.130 | 0.637 | 0.627| 0.647| 0.786 | 0.774| 0.798| 0.364| 0.346 | 0.388 | 0.415 | 0.402| 0.431|
> | **GRPO-7B-n=4** | 0.284| 0.274 | 0.293 | 0.109| 0.102 | 0.117 | 0.681 | 0.671| 0.690| 0.798 | 0.786| 0.807| 0.360| 0.340 | 0.385 | 0.425 | 0.410| 0.439|
> | **TBRM-7B-n=4** | 0.312| 0.302 | 0.321 | 0.138| 0.128 | 0.145 | 0.685 | 0.675| 0.695| 0.798 | 0.787| 0.808| 0.368| 0.347 | 0.388 | 0.437 | 0.423| 0.451|
>
>
> **Q: Relation between TBRM Loss and L2 Loss of AWR**
>
> In our Eq.3, logits are effectively Q-function, and $\pi_\theta$ are effectively the softmax policy of logits/Q-function. We will then explain the difference from the $L_2$ version of AWR based on this.
>
> If our understanding is correct, the $L_2$ version of AWR is doing Monte Carlo style regression, regressing the on-policy log-probability of soft-max policy of return-to-go, namely $\log\mathsf{softmax}\circ R_{\rm rtg}(s ,a)=\log \frac{\exp\{R_{\rm rtg}(s ,a)\}}{Z(s)}$, to $\log\pi(a \mid s)$, where $Z(s)$ is the partition function. In this way, one must do the regression per state-action, which requires to handle the normalization factor $Z(s)$ per state-action.
>
> On the other hand, TBRM is closer to the TD-style algorithm, which is naturally off-policy. TBRM loss is in the trajectory level, and the $\log\pi_\theta$ terms in the middle plays the same role of bootstrap in as in typical TD approaches, which fixes the off-policy data.
>
> We suspect that the reviewer might also question if $L_2$ version of AWR in KL-regularized RL would close that gap. It still cannot, because in that case 1) $L_2$ version of AWR should use $\log\pi_{t-1}$ (log-prob of the policy that collected the data) over return-to-go, i.e., $R_{\rm rtg}(s_h,a_h)=\sum_{h'=h}^H \left( r(s_{h'},a_{h'})-\beta \log\frac{\pi_{t-1}(a_{h'}\mid s_{h'})}{\pi_{\mathsf{ref}}(a_{h'}\mid s_{h'})}\right)$, but TBRM should use $\log\pi_\theta$ (log-prob of current policy), i.e., $R_{\rm rtg}(s_h,a_h)=\sum_{h'=h}^H \left( r(s_{h'},a_{h'})-\beta \log\frac{\pi_\theta(a_{h'}\mid s_{h'})}{\pi_{\mathsf{ref}}(a_{h'}\mid s_{h'})}\right)$. They are same in purely on-policy but not for off-policy; 2) $L_2$ version of AWR should still have squared loss per-state-action, but TBRM has trajectory-level loss.
>
> **Q: Extension to Non-Mathematical Tasks**
>
> To demonstrate the generalizability of our method beyond mathematical tasks, we evaluate it on five tasks from the reasoning-gym [2] under the "graphs" category: course_schedule, family_relationships, largest_island, quantum_lock, and shortest_path. These tasks are naturally represented as graphs, consisting of nodes and edges, and typically require traversing connections to identify relationships, compute optimal paths, or determine reachable components. They involve reasoning patterns that differ significantly from those in mathematical tasks.
>
> We construct a training set of 10,000 problems, with 2,000 questions per task, and a test set of 500 problems, comprising 100 questions from each task. For both training and evaluation, we use the official verifiers provided by reasoning-gym to compute rewards. Our experiments are conducted on Qwen2.5-Math-1.5B using both TBRM and GRPO, with a prompt batch size of 1024 and 4 sampled responses per question (n = 4). Models are trained for 100 steps. All evaluations are conducted using greedy decoding. Results demonstrate that TBRM generalizes well to diverse reasoning tasks and performs on par with GRPO.
>
> | | course_schedule | family_relationships | largest_island | quantum_lock | shortest_path | Average |
> |-|-|-|-|-|-|-|
> | Qwen2.5-Math-1.5B | 29.5% | 3.0%| 11.0%| 5.5% | 0.0% | 9.8%|
> | GRPO (n=4) | 54.0% | 84.0% | 34.0%| 30.5%| 26.0%| 45.7% |
> | TBRM (n=4) | 60.0% | 80.0% | 38.0%| 27.0%| 31.0%| 47.2% |
>
> ---
>
> [1] From Crowdsourced Data to High-Quality Benchmarks: Arena-Hard and BenchBuilder Pipeline. Li et al. 2024
>
> [2] REASONING GYM: Reasoning Environments for Reinforcement Learning with Verifiable Rewards. Stojanovski et al. 2025

---

> > ### Comment · Reviewer_nij4 · 2025-08-06
> >
> > I thank the authors for their detailed response. I believe the additional experiments and explanations will strengthen the paper. However, I still have some concerns about the links between the theoretical insights and strong experimental results. Therefore, I am keeping my score.

---

### Note · Authors · 2025-08-16

We present TBRM, an algorithm that naturally adapts classical BRM to LLMs, yielding a simple yet effective off-policy algorithm using the model's own logits as Q-values. We prove finite-sample convergence to the near-optimal KL-regularized policy using arbitrary off-policy data via an improved change-of-trajectory-measure analysis. Our results suggest that value-based RL is a principled and efficient way to enhance LLM reasoning.

We substantially expanded our experiments and analyses:

**TBRM vs. GRPO** (ufKS and nMNz)\
TBRM is simpler than GRPO since it requires neither importance-sampling ratios nor clipping. It is also more flexible, supporting off-policy training, working with a single rollout, and providing finite-sample guarantees. These strengths make it a practical alternative, especially for multi-turn conversations and agentic workflows where trajectory collection is costly.

**TBRM vs. Related Algorithms** (nij4, 3Y8d, and nMNz)\
We analyze TBRM in relation to other algorithms. Compared to the L2 loss of AWR [1], TBRM requires no return-to-go partition function and operates at the trajectory level. Moreover, its gradient decomposes into two parts: REINFORCE gradient with a baseline applied to off-policy data and a value function regression term, distinguishing it from standard policy gradient methods. Additional comparisons are provided in the updated Appendix E of the supplementary material.

**Extension to Tasks of Other Domains** (nij4 and ufKS)\
We extend our experiments beyond mathematical tasks by evaluating TBRM on the graph category of Reasoning-Gym [2]. The results show that TBRM generalizes well to this broader class of reasoning tasks and achieves performance on par with GRPO.

We also provide several additional results. Specifically, we report confidence intervals for evaluation results (nij4), analyze LLM rollout modeling under MDPs vs. contextual bandits (3Y8d), conduct an ablation on state-action-level BRM (nMNz), and rerun large-scale experiments (3Y8d, nMNz). These details can be found in the corresponding reviewer responses above.

We sincerely appreciate the time and effort the reviewers dedicated to reviewing our work, and we hope that our responses have addressed their concerns.

---
[1] Advantage-weighted regression: Simple and scalable off-policy reinforcement learning. Peng, Xue Bin, et al. 2019\
[2] REASONING GYM: Reasoning Environments for Reinforcement Learning with Verifiable Rewards. Stojanovski et al. 2025

---

### Decision · Program_Chairs · 2025-09-17

**Decision:**

Accept (poster)

**Comment:**

The reviewers find this work to be an interesting contribution, proposing TBRM, a novel Q-learning approach for the reinforcement learning fine-tuning of LLMs. They commend the significance of the core technical idea, the well-written presentation, and the good empirical setup which demonstrates reasonable improvements over existing baselines. While the reviewers note a major weakness in the paper's current lack of comparison to closely related loss functions in prior literature, they believe the work is already just over the bar for acceptance.

While acceptance is recommended for this paper. The authors are still advised to address issues in the final paper, e.g., by explicitly detailing how their proposed loss relates to and differs from established methods like soft Q-learning and policy gradient with a KL penalty.